# Biomaterial-based scaffold for in situ chemo-immunotherapy to treat poorly immunogenic tumors

Hua Wang [1,2,4], Alexander J. Najibi [1,2,4], Miguel C. Sobral[1,2], Bo Ri Seo[1,2], Jun Yong Lee[1,2,3], David Wu [1,2], Aileen Weiwei Li[1,2], Catia S. Verbeke[1,2] & David J. Mooney [1,2✉]

Poorly immunogenic tumors, including triple negative breast cancers (TNBCs), remain resistant to current immunotherapies, due in part to the difficulty of reprogramming the highly immunosuppressive tumor microenvironment (TME). Here we show that peritumorally injected, macroporous alginate gels loaded with granulocyte-macrophage colony-stimulating factor (GM-CSF) for concentrating dendritic cells (DCs), CpG oligonucleotides, and a doxorubicin-iRGD conjugate enhance the immunogenic death of tumor cells, increase systemic tumor-specific CD8 + T cells, repolarize tumor-associated macrophages towards an inflammatory M1-like phenotype, and significantly improve antitumor efficacy against poorly immunogenic TNBCs. This system also prevents tumor recurrence after surgical resection and results in 100% metastasis-free survival upon re-challenge. This chemo-immunotherapy that concentrates DCs to present endogenous tumor antigens generated in situ may broadly serve as a facile platform to modulate the suppressive TME, and enable in situ personalized cancer vaccination.

[1] Harvard John A. Paulson School of Engineering and Applied Sciences, Harvard University, Cambridge, Massachusetts 02138, USA. [2] Wyss Institute for Biologically Inspired Engineering, Cambridge, Massachusetts 02138, USA. [3] Department of Plastic and Reconstructive Surgery, College of Medicine, The Catholic University of Korea, Seoul, Republic of Korea. [4] These authors contributed equally: Hua Wang, Alexander J. Najibi. ✉email: mooneyd@seas.harvard.edu

Cancer immunotherapies that stimulate the patient's own immune system to combat cancers have achieved dramatic progress in the past decade[1,2]. Among them, immune checkpoint blockade therapies such as anti-CTLA-4 and anti-PD-1 have demonstrated clinical success in multiple types of cancers[3,4]. However, this category of cancer immunotherapy is still limited by low patient response rates and potentially severe side effects[5,6]. Cancer vaccines that deliver antigens and adjuvants to antigen-presenting cells (APCs) (e.g., dendritic cells (DCs)) and subsequently elicit antigen-specific cytotoxic T lymphocyte (CTL) and humoral responses can synergize with these cancer immunotherapies to enhance response rates and potentially reduce adverse effects[7,8]. However, traditional cancer vaccines that deliver mixtures of antigens and adjuvants are not effective in treating established tumors[9]. A variety of DC, nanomaterial, and biomaterial scaffold-based cancer vaccines have been developed to improve antigen-specific CTL and humoral responses, and have demonstrated potential in clinical trials[10–15]. Scaffold-based vaccines have utilized poly-lactide-co-glycolide matrices, cryogels, mesoporous silica rods, and pore-forming alginate gels to gradually release granulocyte-macrophage colony-stimulating factor (GM-CSF) to concentrate and educate DCs with synthetic tumor antigens at a site of vaccination[12,16–18]. The injectable pore-forming alginate gel, composed of a bulk, nanoporous alginate network, and rapidly hydrolyzing alginate porogen beads, involves simple preparation, minimally invasive injectable delivery, and ready incorporation of a wide array of therapeutics[18,19]. However, significant challenges still limit broad clinical efficacy with cancer vaccines, including transient antigen-specific CTL responses, lack of tumor-infiltrating lymphocytes, and limited availability of tumor-associated antigens (TAAs).

Vaccine targeting of patient-specific TAAs presents a promising strategy to generate patient-specific, potent immune responses. Recently developed strategies to identify neoantigens have fueled this approach and demonstrated its potential[20,21]. However, the neoantigen strategy requires a substantial delay before patient treatment to allow for tumor sampling and sequencing, prediction of immunogenic mutations, and synthesis of antigens before incorporation into a vaccine. This approach

also requires patient-specific manufacturing of each vaccine, which creates significant financial and technical complexity[22]. In contrast, it has been widely recognized that APCs, including DCs, in the tumor microenvironment (TME) can sample TAAs from dying tumor cells[23,24]. Conventional chemotherapy can directly kill tumor cells and facilitate the release of TAAs in the TME[25]. Certain chemotherapeutic drugs, such as anthracyclines, are reported to induce immunogenic death of tumor cells and facilitate antitumor immune responses[26–28]. However, the resulting antitumor efficacy has been modest, especially in solid tumors[27].

Notably, triple-negative breast cancer (TNBC) represents a class of poorly immunogenic tumors associated with low survival rates and high resistance to current immunotherapies. TNBC is characterized by the lack of estrogen receptor, progesterone receptor and Her2/neu, and patients are typically treated with neoadjuvant chemotherapy followed by surgical removal of tumors and subsequent long-term chemotherapy or radiation therapy[29]. Nevertheless, recurrence and metastasis rates are high, and patients suffer from severe long-term side effects[30,31]. Therefore, a therapy that can effectively control growth and prevent the recurrence and metastasis of TNBCs with minimal off-target side effects is desirable.

Here we propose that immunogenic chemotherapy can be used in concert with a biomaterial that concentrates DCs at a tumor site to sample and process TAAs from dying tumor cells and amplify antitumor CTL responses (Fig. 1). This strategy bypasses the need to identify and deliver antigens in a cancer vaccine, and we aim to explore its potential in treating TNBC. In this approach, a peritumorally injected pore-forming alginate gel loaded with GM-CSF and a tumor-penetrating doxorubicin-iRGD conjugate (Dox-iRGD) locally releases these drugs, inducing immunogenic death of TNBC cells while concentrating a large number of DCs to process endogenous tumor antigens in situ (Fig. 1). The DCs can sample antigens while they migrate towards and into the gel scaffold, where they are activated. Incorporating the TLR9 agonist CpG as an adjuvant enhances DC activation, TME inflammation, and tumor-specific CTL responses, extending mouse survival.

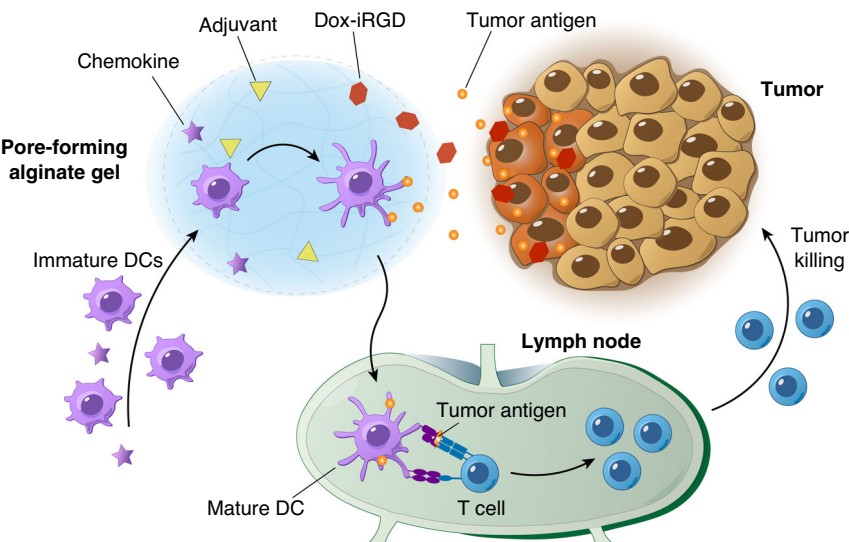

**Fig. 1 Schematic illustration of the in situ cancer vaccine composed of a biomaterial scaffold (gray circle) loaded with chemokines, adjuvants, and chemotherapeutic drugs (Dox-iRGD).** The biomaterial is injected peritumorally, and Dox-iRGD is released to penetrate into tumors and induce immunogenic death of tumor cells, whereas released chemokines can accumulate large numbers of immature dendritic cells (DCs) at the scaffold site. Accumulated DCs can take up and process tumor antigens while being activated with adjuvants to prime tumor-specific T cells for tumor cell killing.

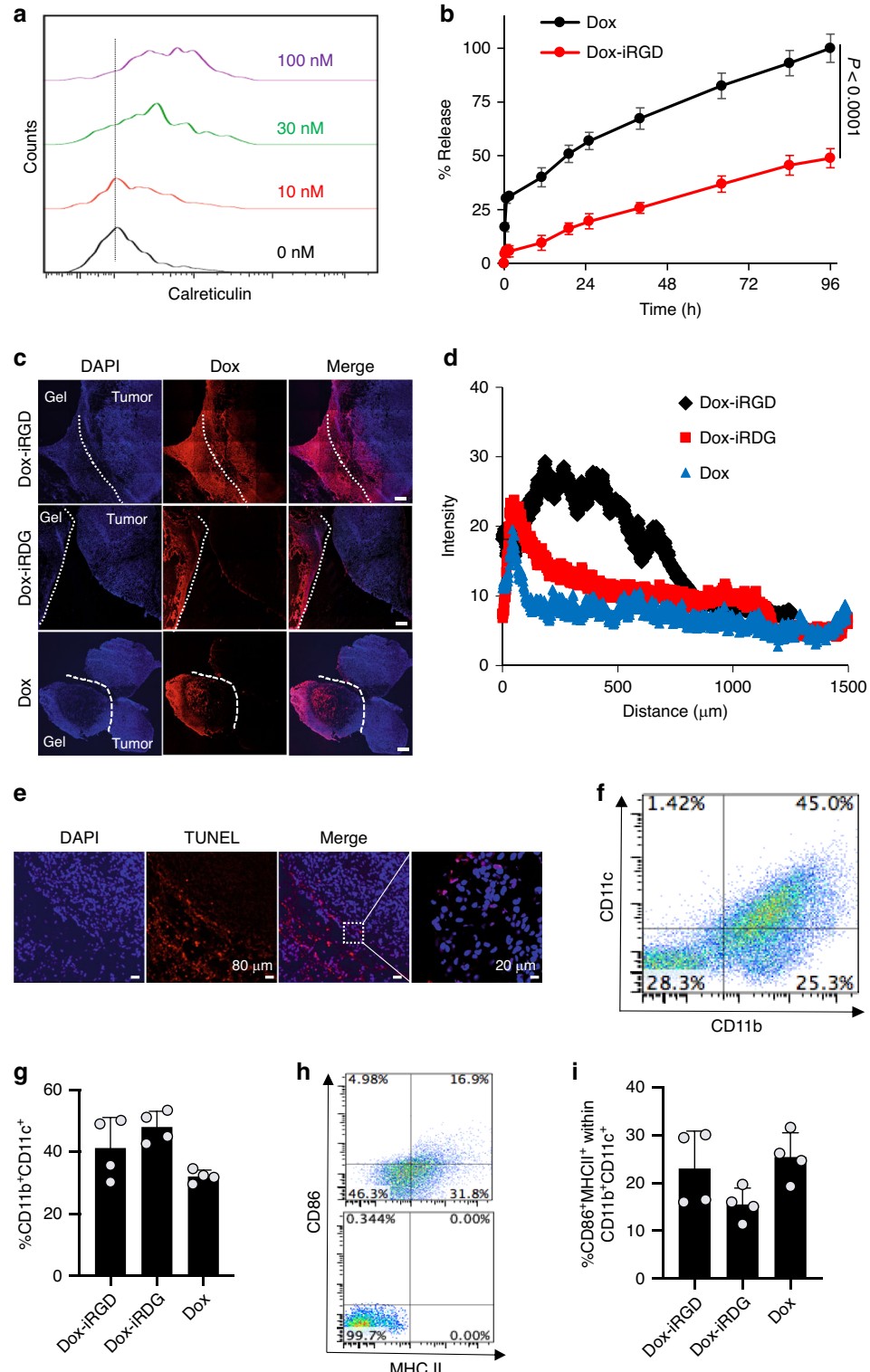

## Results

**Gel scaffold recruits DCs and releases Dox-iRGD into TNBC.**
We first studied whether an anthracycline commonly used in the clinic, doxorubicin (Dox), can induce the immunogenic death of 4T1 TNBC cells in vitro and in vivo. 4T1 cells cultured with Dox for 24 h upregulated cell-surface calreticulin, which has been reported as a biomarker for the immunogenic death of cancer cells[26,27,32]. Calreticulin surface expression increased with Dox concentration in the range of 0–100 nM (Fig. 2a and

Supplementary Fig. 1a). 4T1 cells treated with Dox also increased expression of the anti-phagocytic marker CD47 (Supplementary Fig. 1b, c), previously demonstrated to aid in immunoevasion by cancer cells[33,34]. To circumvent the rapid clearance, limited tumor accumulation, and severe off-target cytotoxicity of systemic Dox delivered intravenously or intraperitoneally[35], we aimed to incorporate Dox into pore-forming alginate gels for controlled and local release to tumors. Pore-forming alginate gels encapsulating Dox were formed by mixing a 3% (w/v) alginate

**Fig. 2 Pore-forming alginate gels containing Dox-iRGD and GM-CSF induce apoptosis of 4T1 tumor cells, and concentrated DCs in situ.**
**a** Representative flow cytometry histograms for calreticulin on the surface of 4T1 cells after treatment with different concentrations of Dox for 24 h in vitro.
**b** Release profiles of Dox and Dox-iRGD from pore-forming alginate gels. $n = 4$ biologically independent samples per group. Statistical analysis was performed using two-tailed $t$-tests. **c–i** Gels containing Dox-iRGD and GM-CSF were peritumorally injected when the tumors reached a diameter of 6–7 mm. This experiment was repeated once. **c** Confocal images of tumor and gel sections at 4 days post peritumoral injection of pore-forming gels containing Dox-iRGD (red), Dox-iRDG (red), and Dox (red), respectively. GM-CSF was incorporated in all groups. Cell nuclei were stained with DAPI (blue). White dotted lines indicate the tumor-gel boundary. Scale bar: 200 μm. **d** Semi-quantitative tumor penetration profiles of Dox-iRGD, Dox-iRDG, and Dox, respectively. **e** Representative TUNEL staining of 4T1 tumors from mice treated with gels containing GM-CSF and Dox-iRGD. Scale bar: 80 μm, except for last image, which is a magnified view of the merged image. This experiment was repeated once. **f** Representative flow cytometry plot of $CD11b^+CD11c^+$ cells in Dox-iRGD loaded gels at 18 days post gel injection. **g** Percentage of $CD11b^+CD11c^+$ DCs among the recruited cells in pore-forming gels. $n = 4$ biologically independent animals per group. **h** Representative flow cytometry plot of $CD86^+MHCI^+$ cells within $CD11b^+CD11c^+$ cells in Dox-iRGD loaded gels at 18 days post gel injection (upper panel). Lower panel contains the plot for isotype controls. **i** Percentage of $CD86^+MHCI^+$ cells within $CD11b^+CD11c^+$ cells in pore-forming gels. $n = 4$ biologically independent animals per group. All the numerical data are presented as mean ± SD. Source data are provided as a Source Data file.

solution containing Dox with degradable porogens before cross-linking with calcium ions. Dox exhibited a rapid release profile from the gels, with ~30% being released within 6 h and more than 60% being released within 24 h (Fig. 2b). To achieve a more sustained release and potentially aid in tumor penetration, Dox was conjugated with a tumor-penetrating iRGD peptide. iRGD (CRGDKGPDC) is a nine-amino acid cyclic peptide that can bind to αVβ3 and αVβ5 integrins, and further bind to neutropilin-1, and has shown good tumor-penetrating capability[36–38]. Briefly, Dox was modified with a maleimide functional group via an acid-labile hydrazone linker, followed by conjugation to thiol-bearing iRGD to yield Dox-iRGD (Supplementary Fig. 2a). Dox-maleimide was characterized by liquid chromatography-mass spectrometry and proton nuclear magnetic resonance ([1]H NMR) (Supplementary Fig. 2b–d), and Dox-iRGD was characterized by [1]H NMR and matrix-assisted laser desorption/ionization mass spectrometry (Supplementary Fig. 2d, e). As expected, Dox-iRGD was rapidly degraded into Dox at pH 4, but remained stable at pH 8 (Supplementary Fig. 2f). Dox-iRGD and a scrambled control Dox-iR**D**G showed similar anticancer effect against 4T1 cells in vitro, with a higher $IC_{50}$ value than Dox (Supplementary Fig. 2g) as expected. Compared to Dox, Dox-iRGD showed a slower release profile from the pore-forming alginate gel, with ~20% being released within 24 h and ~28% being released within 48 h in vitro, presumably due to greater number of positive charges and increased electrostatic interactions between Dox-iRGD and alginate gels, as compared to Dox (Fig. 2b).

We next studied whether Dox released from peritumorally injected pore-forming gels can penetrate into 4T1 tumors and induce the apoptosis of tumor cells. Balb/c mice bearing subcutaneous 4T1 tumors (6–7 mm in diameter) were peritumo-rally injected with gels containing GM-CSF along with either Dox-iRGD, Dox-iR**D**G, or unmodified Dox, and gels and tumors were collected for analyses 4 days later. Confocal imaging of gel and tumor sections showed significantly improved tumor penetration of Dox-iRGD compared to Dox-iR**D**G (Fig. 2c, d), substantiating the previously reported tumor-penetrating capacity of iRGD[39,40]. Unmodified Dox showed negligible accumulation in tumor tissues (Fig. 2c, d), presumably due to its rapid release from gels and poor retention in surrounding tissues. Within the penetration range, Dox-iRGD induced significant apoptosis of tumor cells (Fig. 2e and Supplementary Fig. 4c). To confirm the ability of these gels to concentrate DCs, gels were collected 18 days post gel injection and infiltrating cells characterized. A significant number of cells (~1.6 million) were found in all gels, and ~40% and 32% were $CD11b^+CD11c^+$ DCs in Dox-iRGD and Dox-loaded gels, respectively (Fig. 2f, g). Approximately 23% of DCs in the Dox-iRGD-loaded gels were $CD86^+MHCII^+$ (Fig. 2h, i).

**Gel vaccine boosts antitumor CTL response and efficacy.** The antitumor efficacy of gels with Dox-iRGD and GM-CSF was next explored. Gels loaded with Dox-iRGD alone did not show any therapeutic benefit against 4T1 tumors (Supplementary Fig. 3). With GM-CSF, Dox-iRGD-loaded gels significantly slowed the growth of 4T1 tumors in comparison to Dox-iRDG or Dox-loaded gels, or untreated animals (Fig. 3a and Supplementary Fig. 4a). Cancer metastases were observed in lungs of all groups at 18 days post gel injection (Fig. 3b), but mice treated with Dox-iRGD-loaded gels displayed significantly fewer pulmonary tumor nodules compared to unmodified Dox or untreated mice (Fig. 3b, c). Both histological evaluation of the spinal bone marrow tissue and body weight change of mice showed higher toxicity of Dox-loaded gels than Dox-iRGD-loaded gels (Fig. 3d and Supplementary Fig. 4b, d), presumably because of more significant leakage of Dox into the bloodstream and resulting systemic exposure. Together, these experiments suggest that peritumorally injected pore-forming gels loaded with Dox-iRGD and GM-CSF can release Dox-iRGD into 4T1 tumors, induce apoptosis of tumor cells, concentrate DCs in situ, and slow tumor growth while reducing off-target systemic toxicity.

Next, we explored the ability of the adjuvant CpG to further potentiate in situ vaccination and antitumor efficacy. After 4T1 tumor inoculation on day 0, gels loaded with Dox-iRGD (100 μg) and CpG (50 μg), CpG (50 μg) alone, or Dox-iRGD (100 μg) alone were injected peritumorally on day 5 (Supplementary Fig. 5a). GM-CSF was incorporated to concentrate DCs in all gel conditions of this and all subsequent studies unless otherwise noted. All three treatment groups showed a reduced tumor growth rate compared to the untreated group (Fig. 4a). Compared to either Dox-iRGD or CpG alone, Dox-iRGD with CpG resulted in significantly slower tumor growth (Fig. 4a). The median survival of mice treated with gels loaded with Dox-iRGD and CpG was significantly longer than the CpG alone group, the Dox-iRGD-alone group, and the untreated group (Fig. 4b and Supplementary Fig. 5b). Taken together, these data suggest that the combination of Dox-iRGD and adjuvant CpG has the potential to improve antitumor efficacy, presumably due to improved activation of recruited DCs, and facilitated antigen presentation and T-cell priming processes. However, lung metastases were still observed in mice treated with Dox-iRGD and CpG at the end of the study, indicating insufficient potency of the generated antitumor CTL responses.

We then investigated whether the antitumor efficacy of the in situ gel vaccine could be further improved by increasing the dose of Dox-iRGD and CpG. After 4T1 tumor inoculation on day 0, gels loaded with Dox-iRGD (200 μg) and CpG (100 μg) were injected next to the tumors on day 5. To characterize the immune response, DC recruitment and activation in the gels

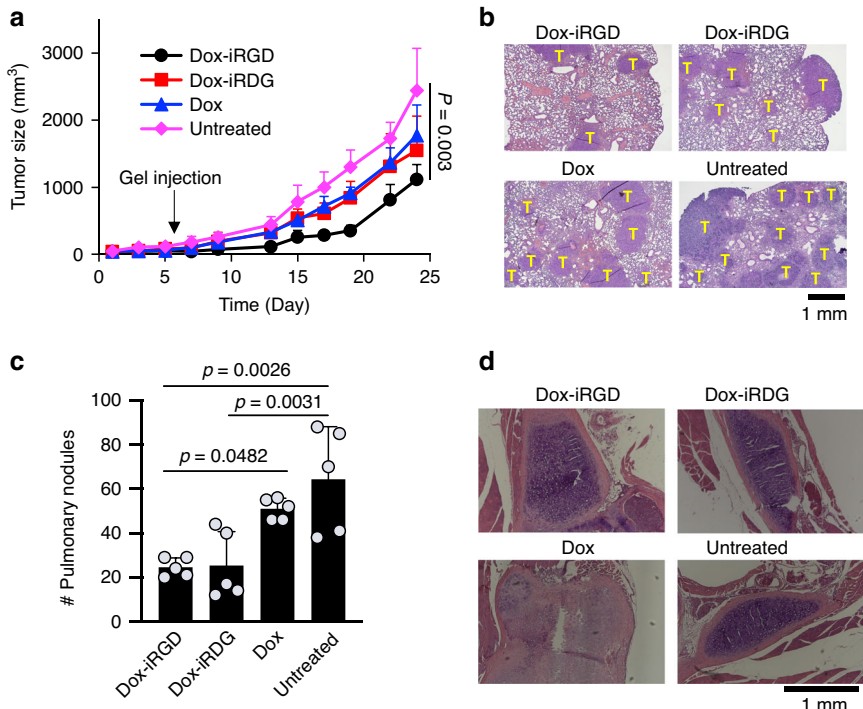

**Fig. 3 Pore-forming alginate gels containing Dox-iRGD and GM-CSF delay 4T1 tumor growth and reduce metastases.** Gels were peritumorally injected when the tumors grew to ~6–7 mm. GM-CSF was incorporated in all gels. Mice without gel treatment were used as controls. **a** Average 4T1 tumor volume of each group over the course of the efficacy study. $n = 5$ biologically independent animals per group. Statistical analysis was performed using ANOVA with Fisher's LSD post hoc test. **b** Representative images of H&E-stained lung tissues collected at 18 days post peritumoral injection of gels containing Dox-iRGD, Dox-iRDG, and Dox, respectively. T indicates tumors. **c** Average tumor nodule counts on the lung tissues at 18 days post gel injection. $n = 5$ biologically independent animals per group. Statistical analysis was performed using ANOVA with Fisher's LSD post hoc test. **d** Histopathology of spinal bone marrow tissues collected at 18 days post gel treatment. Scale bar: 1 mm. For **b** and **d**, experiments were repeated once. All the numerical data are presented as mean ± SD, except for **a** where mean ± SEM is used. Source data are provided as a Source Data file.

were analyzed, as was the presence of 4T1-specific CD8$^+$ T cells in spleens and tumor-draining lymph nodes (tdLNs) 4 days post gel injection. It is noteworthy that cells in the gels that were harvested at different times showed high viability (>85%; Supplementary Fig. 6). Gels loaded with Dox-iRGD and CpG recruited a similar total number of cells and CD11c$^+$ DCs, compared to gels loaded with Dox-iRGD alone (Supplementary Fig. 7a, b). However, more DCs in the former group expressed an activated phenotype, with 28% CD86$^+$MHCII$^+$ DCs (Supplementary Fig. 7c, d). Strikingly, 14% of CD8$^+$ T cells from tdLNs of mice treated with gels containing Dox-iRGD and CpG were IFN-γ$^+$ upon restimulation by 4T1 cells ex vivo, indicating 4T1 antigen specificity, which is significantly higher than found from mice treated with gels containing Dox-iRGD alone (5.3%) or untreated mice (2.4%) (Fig. 4c–e). CD8$^+$ T cells isolated from the spleens and CD4$^+$ T cells isolated from spleens/tdLNs of mice treated with gels containing Dox-iRGD and CpG also showed increased expression of interferon-γ (IFN-γ) in response to 4T1 cells, as compared to Dox-iRGD alone or untreated groups (Supplementary Fig. 8). Similarly, at 11 days post gel injection, significantly increased numbers of 4T1-specific CD8$^+$ and CD4$^+$ T cells in tdLNs and spleens were also detected in mice treated with gels containing Dox-iRGD and CpG, in comparison to mice treated with gels containing Dox-iRGD alone, mice treated with gels containing CpG alone, or untreated mice (Supplementary Fig. 9). The observed DC activation and IFN-γ$^+$ T-cell responses in mice treated with gels containing the increased dose of Dox-iRGD and CpG correlated with significantly reduced tumor growth compared to untreated mice (Fig. 4f and Supplementary

Fig. 10a, b), with median survival increasing by 50% to 45 days (Fig. 4g and Supplementary Fig. 10c). By giving a booster gel vaccine on day 12 (7 days after the first gel injection), tumor control was further improved, with a median survival of 49 days (Fig. 4f, g and Supplementary Fig. 10c). Although tumor burden was low in gel vaccine groups at study termination, lung metastases were observed in mice, and these were likely responsible for animal loss. Histological evaluation of liver, heart, spleen, and kidney tissues showed minimal off-target toxicity of the higher-dose in situ gel vaccine compared to untreated mice (Supplementary Fig. 10d). To study whether the recruited DCs in gel scaffolds could sample tumor antigens, the gel vaccine was injected adjacent to subcutaneous green fluorescent protein (GFP)-expressing 4T1 tumors (Supplementary Fig. 11a). At 4 days post injection, a significantly higher percentage of GFP-positive DCs was observed in the gel vaccine group compared to control gels without incorporation of Dox-iRGD (Supplementary Fig. 11b, c). DCs in tdLNs also showed a stronger GFP signal in mice treated with the gel vaccine, in comparison to mice treated with control gels without incorporation of Dox-iRGD or untreated mice (Supplementary Fig. 11d–f). It is noteworthy that the recruited DCs tend to leave the gel scaffold when GM-CSF is mostly consumed, and CpG can further upregulate CCR7 receptors of DCs and facilitate their migration from the gel to lymph nodes[12]. Taken together, these experiments demonstrate that the in situ gel vaccine loaded with GM-CSF, Dox-iRGD, and CpG enhances tumor-specific CD8$^+$ and CD4$^+$ T-cell activity in tdLNs and the systemic circulation, and slows the growth of 4T1 tumors.

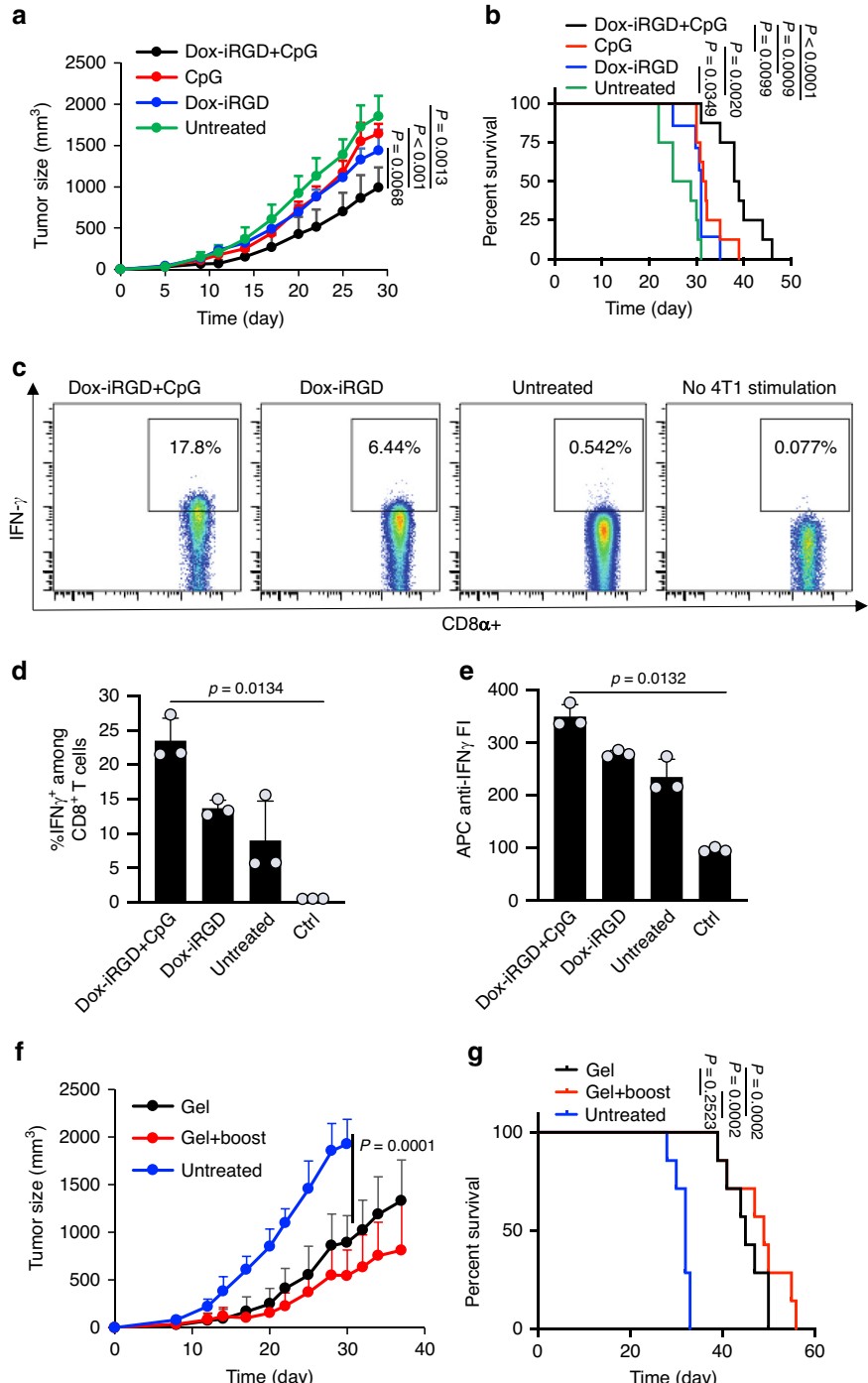

**Fig. 4 Pore-forming gels loaded with GM-CSF, Dox-iRGD, and CpG improve tumor-specific CTL responses and antitumor efficacy against 4T1 triple-negative breast cancer. a**, **b** 4T1 cells were injected subcutaneously on day 0. Mice were untreated or treated with gels containing Dox-iRGD (100 µg) and CpG (50 µg) or Dox-iRGD (100 µg) alone or CpG (50 µg) alone on day 5. GM-CSF was incorporated in all groups. **a** Average 4T1 tumor volume of each group over the course of the efficacy study. $n = 8$ biologically independent animals per group. Statistical analysis was performed using ANOVA with Fisher's LSD post hoc test. **b** Kaplan–Meier plots for all groups. $n = 5$ biologically independent animals per group. Statistical analysis was performed using the log-rank (Mantel–Cox) test. **c**–**g** Following 4T1 tumor inoculation on day 0, gels containing Dox-iRGD (200 µg) and CpG (100 µg) or Dox-iRGD (200 µg) alone or CpG (100 µg) alone were injected peritumorally on day 5. **c** Representative FACS plots of cells isolated from tumor-draining lymph nodes at 4 days post gel injection and restimulated with 4T1 cells. APC-conjugated anti-IFN-γ and Pacific Blue-conjugated anti-CD8 were used for staining. Cells without restimulation were used as control. **d** Percentage of IFN-γ$^+$CD8$^+$ cells. **e** Mean APC fluorescence intensity of CD8$^+$ T cells. For **d**, **e**, $n = 5$ biologically independent animals per group. Statistical analysis was performed using ANOVA with Fisher's LSD post hoc test. **f** Average 4T1 tumor volume of each group over the course of the efficacy study. Statistical analysis was performed using ANOVA with Fisher's LSD post hoc test. **g** Kaplan–Meier plots for all groups. Statistical analysis was performed using the log-rank (Mantel–Cox) test. For **f**, **g**, $n = 7$ biologically independent animals per group. All the numerical data are presented as mean ± SD except for **a** where mean ± SEM is used. Source data are provided as a Source Data file.

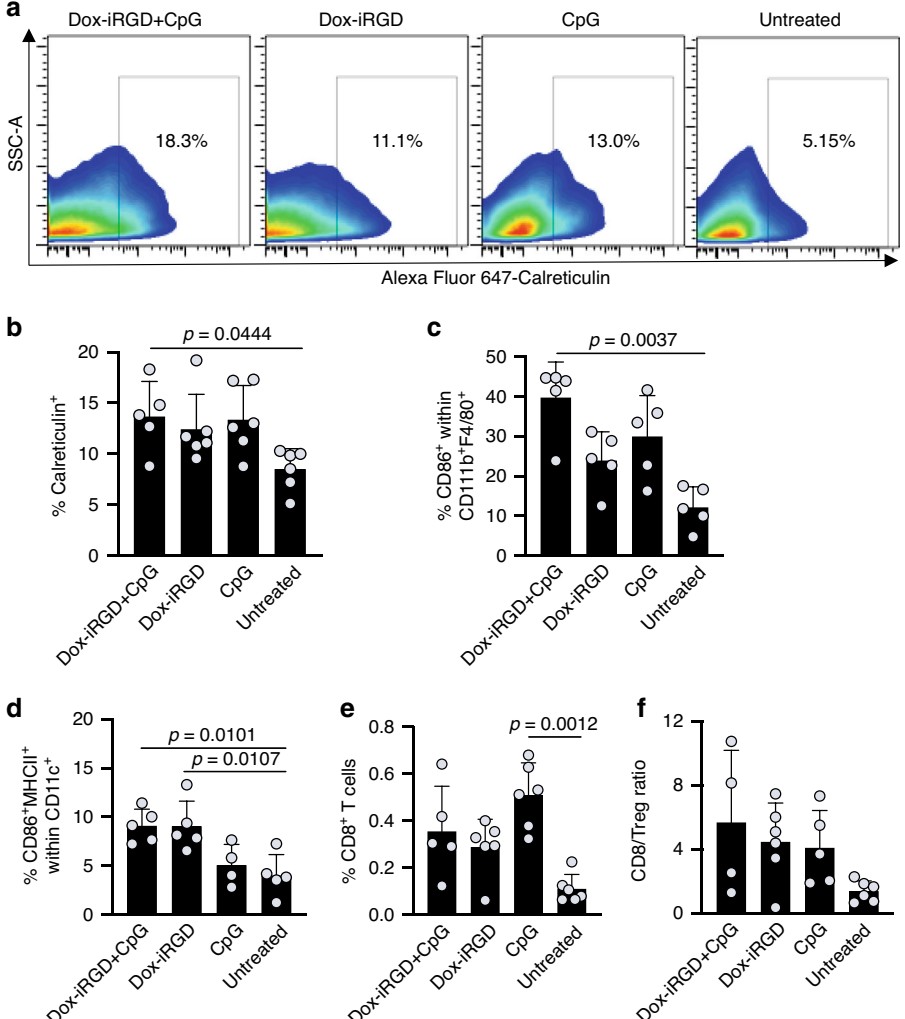

**Fig. 5 Pore-forming gels loaded with GM-CSF, Dox-iRGD, and CpG induce immunogenic death of tumor cells, inflame tumor microenvironment, and increase effector T-cell infiltration.** Following 4T1 tumor inoculation on day 0, gels containing Dox-iRGD (200 μg) and CpG (100 μg), or Dox-iRGD (200 μg) alone or CpG (100 μg) alone were injected peritumorally on day 5. Tumors were analyzed on day 16. **a** Representative FACS plots of tumor cells stained for the immunogenic death marker, calreticulin. CD11b⁺, CD3⁺, and CD8⁺ cells were excluded for these plots. **b** Percentage of calreticulin⁺ tumor cells in different groups. **c** Percentage of CD86⁺ cells among CD11b⁺F4/80⁺ tumor-associated macrophages. **d** Percentage of CD86⁺MHCII⁺ cells among CD11c⁺ DCs in tumors. **e** Intratumoral infiltration of CD8⁺ T cells in different groups. **f** Ratio of CD8⁺ to regulatory T cells in tumors. For **b**–**f**, data are presented as mean ± SD; $n = 6$ biologically independent animals per group. Statistical analysis was performed using ANOVA with Fisher's LSD post hoc test. Source data are provided as a Source Data file.

**Gel vaccine reshapes the TME.** The immunosuppressive TME may constitute a barrier against tumor control, even with strong systemic CTL responses[41]. To better understand the underlying immunomodulatory effects of the in situ gel vaccine, we next analyzed post-treatment changes in the TME, including immunogenic death of tumor cells, population and activation status of tumor-infiltrating DCs, effector T cells and exhaustion, regulatory T (Treg) cells, and tumor-associated macrophages (TAMs). GM-CSF containing gels loaded with both Dox-iRGD and CpG, or either Dox-iRGD or CpG alone were again injected adjacent to 4T1 tumors. Gels and tumors were harvested for analyses 11 days later, as the in situ vaccine had shown efficacy in slowing tumor growth at this time. Cells in tumors treated with gels containing Dox-iRGD and CpG showed enhanced cell-surface expression of calreticulin compared to tumors treated with gels containing Dox-iRGD alone or untreated tumors, indicating increased immunogenic death of tumor cells (Fig. 5a, b). Tumors treated with gels containing CpG alone also showed an increased expression of calreticulin in comparison to untreated tumors

(Fig. 5a, b). The cell-surface expression of CD47 was also enhanced in tumors treated with gels containing Dox-iRGD and CpG compared to untreated tumors (Supplementary Fig. 12a, b), presumably as part of the immune evasion mechanism of tumor cells in response to the boosted immune responses. We also examined the population of M1/M2 macrophages in the TME. Although the definitive markers of M1 and M2 macrophages remain debatable, CD86, CD163, and CD206 gating within CD11b⁺F4/80⁺ cells has been widely used to evaluate M1/M2 polarization[42–45]. The number of CD11b⁺F4/80⁺ TAMs in the TME was not significantly changed after the gel treatment (Supplementary Fig. 12c). However, mice treated with gels containing Dox-iRGD and CpG showed a significantly increased number of CD86⁺ TAMs, known as pro-inflammatory M1-type macrophages, compared to mice treated with gels containing Dox-iRGD alone or untreated mice (Fig. 5c and Supplementary Fig. 12d). Together, these data suggest that the in situ gel vaccine is able to repolarize TAMs towards the pro-inflammatory M1-phenotype, which has been shown to be positively associated with

improved anticancer efficacy and extended survival in both pre-clinical and clinical settings[46].

Gels incorporating Dox-iRGD with or without CpG roughly increased the number of activated (CD86+MHCII+) tumor-infiltrating CD11c+ DCs by 50% (Fig. 5d and Supplementary Fig. 13b–d) as compared to treatment with gels containing CpG alone or untreated tumors, although the total number of DCs remained unchanged (Supplementary Fig. 13a). Tumor infiltration of CD8+ T cells was also significantly improved in all three treatment groups (Fig. 5e and Supplementary Fig. 14a), along with a significantly increased CD8+/FoxP3+ Treg cell ratio (Fig. 5f and Supplementary Fig. 14b). Tumors treated with gels containing CpG alone showed a slightly increased infiltration of CD8+ T cells as compared to tumors treated with gels containing Dox-iRGD and CpG, but the number of Treg cells also increased, resulting in a slightly lower CD8+/Treg ratio (Fig. 5f and Supplementary Fig. 14b). Finally, as the antitumor efficacy of vaccines can be dampened by upregulation of homeostatic immune checkpoint pathways, expression of classical immunor-egulatory and exhaustion markers on tumor-infiltrating CD8+ T cells was next analyzed. Approximately 30% of CD8+ T cells in tumors treated with Dox-iRGD and CpG were PD-1-positive, with no significant differences across the different treatment groups (Supplementary Fig. 15a). TIM3 and LAG3 expression were also found on a subset of intratumoral CD8+ T cells in all groups and no significant differences between conditions were observed (Supplementary Fig. 15b–d). Interestingly, in contrast, the PD-L1 expression on the surface of tumor cells was significantly upregulated in the treatment groups. Specifically, 13% of tumor cells treated with gels containing Dox-iRGD and CpG were PD-L1-positive, as compared to 3% in untreated tumor cells (Fig. 6a, b). Tumors treated with gels containing CpG alone exhibited a similar PD-L1 expression as tumors treated with gels containing Dox-iRGD and CpG, both were significantly higher than tumors treated with gels containing Dox-iRGD alone (Fig. 6a, b). Together, these data indicate that the in situ vaccine improves the activation of tumor-infiltrating DCs and tumoral infiltration of CD8+ T cells, increases CD8+/Treg number ratios, and upregulates PD-L1 expression of tumor cells.

**Gel vaccine synergizes with checkpoint blockade therapy.** Considering the upregulated PD-L1 expression of tumor cells after treatment with the in situ gel vaccine and intrinsically high PD-1 expression of tumor-infiltrating CD8+ T cells, we next studied whether the antitumor efficacy of the gel vaccines could be further improved via combination with anti-PD-1 checkpoint blockade therapy (Fig. 6c). The gel vaccine group again significantly slowed growth of 4T1 tumors compared to the untreated group (Fig. 6d and Supplementary Fig. 16a). Anti-PD-1 alone did not demonstrate an antitumor effect until day 21, when a slightly decreased tumor growth rate was observed (Fig. 6d and Supplementary Fig. 16a). Mice treated with the combination of gel vaccine and anti-PD-1 showed significantly reduced tumor size compared to mice treated with the gel vaccine alone or anti-PD-1 alone (Fig. 6d and Supplementary Fig. 16a). The median survival of mice treated with anti-PD-1 alone was 28 days, similar to that of untreated mice (27 days) (Fig. 6e and Supplementary Fig. 16b). Mice treated with the combination of gel vaccine and anti-PD-1 or the gel vaccine alone showed significantly longer survival than untreated mice or mice treated with anti-PD-1 alone, with a median survival of 40 and 37.5 days, respectively (Fig. 6e and Supplementary Fig. 16b). Strikingly, mice treated with both gel vaccine and anti-PD-1 developed significantly fewer pulmonary tumor nodules, in comparison to mice treated with anti-PD-1 alone or untreated mice (Fig. 6f). Mice treated with

anti-PD-1 alone also had fewer pulmonary tumor nodules than untreated mice, despite the absence of survival improvement (Fig. 6f). Collectively, these experiments demonstrated that the combination of in situ gel vaccines and anti-PD-1 can better control the development of lung metastases and the growth of primary 4T1 tumors. Future additional optimization of the gel vaccine and optimization of the combination of gel vaccines and checkpoint blockades (e.g., timing and antibody choice) may potentially generate more significant synergy.

**Gel vaccine restrains postoperative tumor recurrence.** To explore a clinically relevant situation, we next studied whether the in situ gel vaccine could prevent cancer recurrence after primary tumor resection. Upon surgical removal of luciferase-expressing 4T1 tumors, a gel vaccine or bolus vaccine (containing all active components of vaccine in phosphate-buffered saline (PBS) without the gel carrier) was injected near the original tumor area (Fig. 7a). Subsequent bioluminescence imaging revealed an increasing tumor signal over time and a high rate of tumor recurrence in the untreated group (Fig. 7b). Both treated groups had significantly lower tumor recurrence, as evidenced by lower bioluminescence signal and improved tumor-free and overall survival (Fig. 7b–d). However, in comparison to the bolus vac-cine, the gel vaccine resulted in significantly slower tumor growth and improved survival (Fig. 7b–d), confirming the importance of a biomaterial scaffold for therapeutic effect. To test for persistent immunological memory, mice were re-challenged with intrave-nously (i.v.) injected luciferase-expressing 4T1 cancer cells (Fig. 7e). Bioluminescence imaging of control mice without gel vaccine treatment confirmed the occurrence and proliferation of 4T1 metastases in the lung parenchyma (Fig. 7f, g). Strikingly, mice previously treated with the gel vaccine showed 100% metastasis-free survival, in contrast to 0% in the untreated group (Fig. 7f–h and Supplementary Fig. 17), demonstrating the potency and persistence of antitumor immune responses generated by the in situ gel vaccine. To rule out the effect of potential immuno-genicity of luciferase antigens, we repeated the tumor resection study using 4T1 cells, and the gel vaccine still resulted in sig-nificantly improved survival (Supplementary Fig. 18a, b). Upon re-challenge with i.v.-injected 4T1 cells, mice previously treated with the gel vaccine still showed 100% metastases-free survival (Supplementary Fig. 18c). Although the ability of Dox-iRGD to penetrate established tumors is likely not relevant in this setting, its sustained, local release and ability to kill residual tumor cells, coupled with in situ concentrated DCs, is likely responsible for the observed antitumor immune responses, as observed in the primary tumor model.

To further evaluate the antitumor efficacy of in situ gel vaccines in an orthotopic tumor model, we injected 4T1 cells into the mammary fat pad, and peritumorally injected the gel vaccines when the tumors grew to ~150 mm$^3$. Similar to previous observations, the in situ gel vaccine was able to significantly slow the tumor growth and improve the survival of mice (Supplementary Fig. 19). We also studied whether the in situ gel vaccine would be effective against other TNBC models. In an orthotopic EMT6 TNBC model, the gel vaccine significantly reduced the tumor growth rate and improved the survival of mice compared to controls, with 30% of tumors being eradicated (Supplementary Fig. 20). These results were mimicked in an orthotopic EO771 TNBC model, where the in situ vaccine again demonstrated therapeutic benefit (Supplementary Fig. 21). As expected, the gel vaccine also showed potent efficacy against 4T07 tumors (Supplementary Fig. 22). It is noteworthy that the gel vaccine did not show any therapeutic benefit in athymic nude mice with a compromised immune system (Supplementary

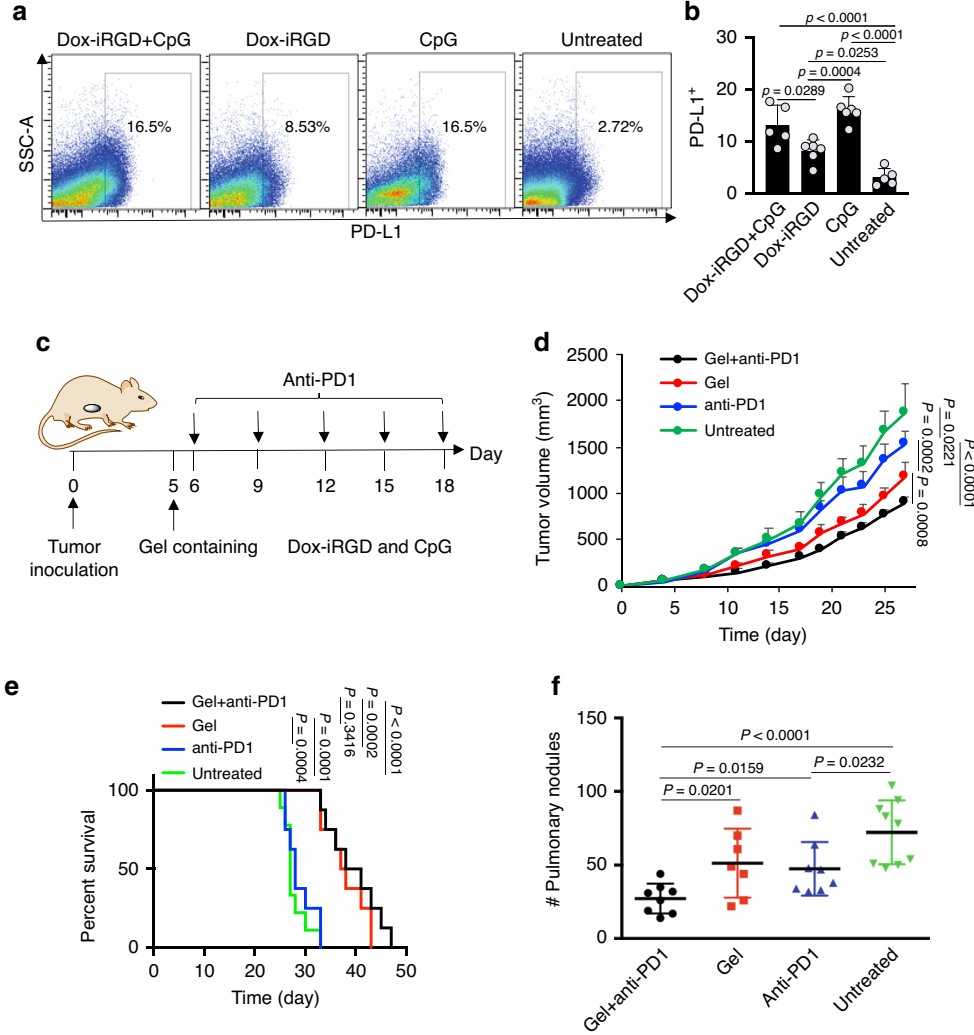

**Fig. 6 In situ gel vaccine combined with anti-PD-1 therapy for tumor control. a** Representative FACS plots for PD-L1 expression of tumor cells (CD11b[+], CD3[+], and CD8[+] cells have been excluded for these plots) and **b** percentage of PD-L1[+] tumor cells at 11 days post treatment with gels containing Dox-iRGD (200 μg) and CpG (100 μg), or Dox-iRGD (200 μg) alone or CpG (100 μg) alone. $n = 6$ biologically independent animals per group. Statistical analysis was performed using ANOVA with Fisher's LSD post hoc test. **c** Time frame of efficacy study. Following 4T1 tumor inoculation on day 0, gels containing Dox-iRGD (200 μg) and CpG (100 μg) were injected next to tumors on day 5, and anti-PD-1 was intraperitoneally injected on days 6, 9, 12, 15, and 18, respectively. **d** Average 4T1 tumor volume of each group over the course of the efficacy study. Statistical analysis was performed using ANOVA with Fisher's LSD post hoc test. **e** Kaplan–Meier plots for all groups. Statistical analysis was performed using the log-rank (Mantel–Cox) test. For **d**, **e**, $n = 9$ biologically independent animals for the untreated group and $n = 8$ for the other three groups. **f** Average tumor nodule counts on the lung tissues harvested from mice at the time of euthanasia. $n = 9$ biologically independent animals for the untreated group, $n = 8$ for the anti-PD-1 and gel + anti-PD-1 groups, and $n = 7$ for the gel group. Statistical analysis was performed using ANOVA with Fisher's LSD post hoc test. All the numerical data are presented as mean ± SD, except for **d** where mean ± SEM is used. Source data are provided as a Source Data file.

Fig. 23). These experiments further validated the potent antitumor efficacy of the in situ gel vaccine against TNBCs.

## Discussion

Although 4T1 tumor growth and metastasis rates varied between experiments in this study, the in situ gel vaccine consistently showed significant therapeutic benefit. It is noteworthy that 4T1 TNBC is a notoriously aggressive tumor model with a high metastasis rate and has proven refractory to various therapies[47–49]. Compared to the chemo- and radiotherapies reported thus far for treating the poorly immunogenic 4T1 tumor model[47,48], the in situ gel vaccine involves one single injection, results in better or comparable efficacy, and reduces systemic toxicity. This single-dose gel vaccine outperforms or is comparative to the vast majority of previously reported immunotherapies in treating 4T1

tumors, including checkpoint blockade therapies, DC vaccines, and immunostimulatory agents[48–52], presumably because of its superior capability to reshape the TME and concentrate DCs to prime potent antitumor T-cell responses. Although combination therapies involving chemotherapeutics and locally administered immunomodulatory agents have previously been explored in this model[53,54], the immune subsets targeted and mechanism of immunomodulation at times remain unclear. In contrast, our system allows for controlled release of rationally designed components with specific functions in a biomaterial niche for concentrating and programming of DCs in situ.

Current clinical treatment of breast cancers typically includes surgery to debulk the tumor, followed by other additional therapies to provide long-term prevention of recurrence/metastasis. In this setting, the in situ gel vaccine also demonstrated significant

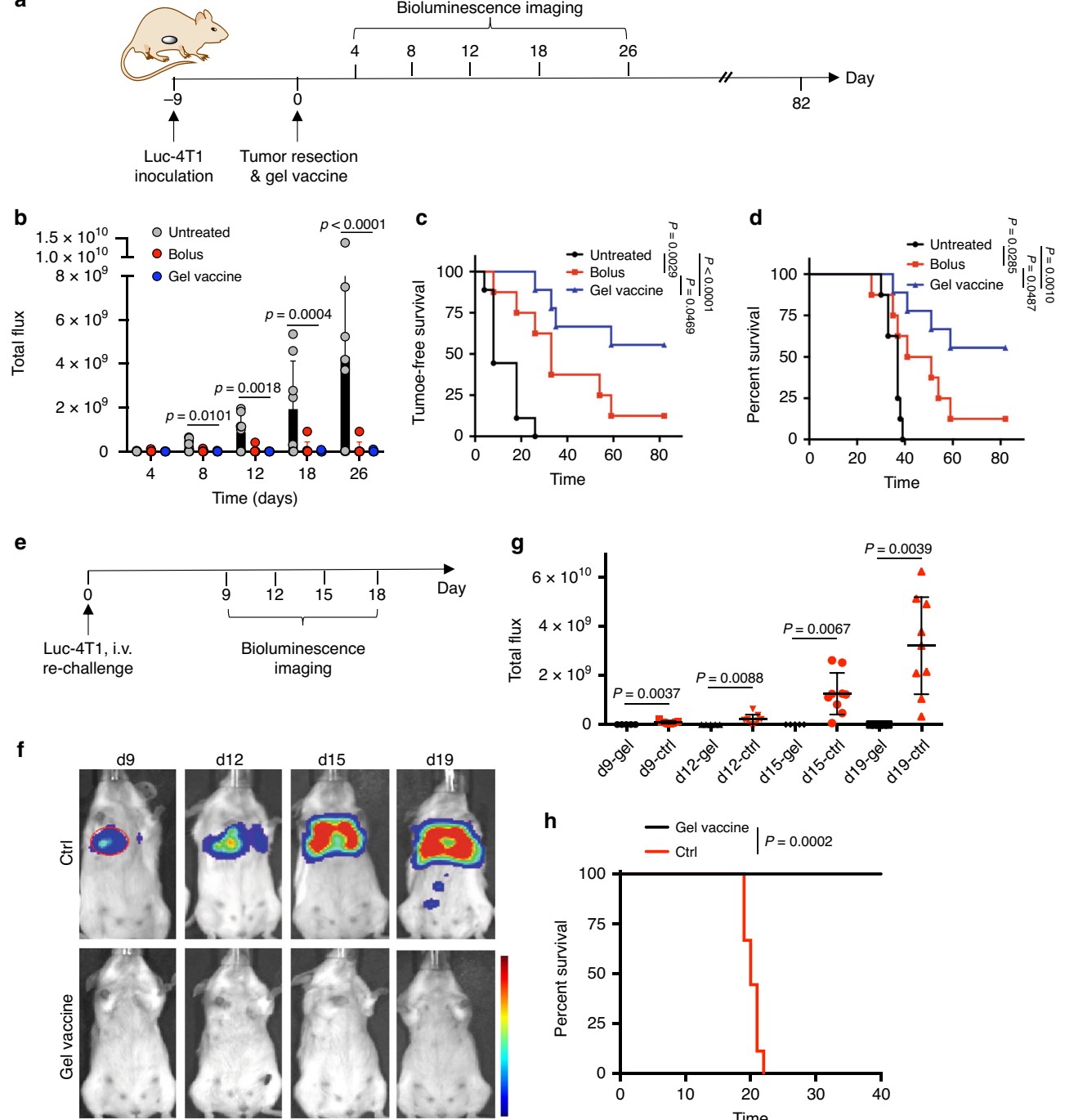

**Fig. 7 In situ gel vaccine prevents tumor recurrence and metastases when applied post-surgical tumor resection. a–d** Following surgical resection of luciferase-expressing 4T1 (luc-4T1) tumors, gels containing GM-CSF, Dox-iRGD (200 μg) and CpG (100 μg) or bolus vaccines (solution of same quantities of GM-CSF, Dox-iRGD, and CpG) were injected at surgical site. **a** Outline of study. **b** Luminescence signals of mice at different times. Statistical analysis was performed using ANOVA with Fisher's LSD post hoc test. **c** Kaplan–Meier plots for tumor-free survival of all groups. **d** Kaplan–Meier plots for overall survival of all groups. For **b**–**d**, $n = 9$ biologically independent animals for gel vaccine group and $n = 8$ for other groups; for **c**, **d**, statistical analysis was performed using the log-rank (Mantel–Cox) test. **e**–**h** Following re-challenge with i.v.-injected luc-4T1 cells at 82 days post surgery, tumor growth and animal survival were monitored. Naive mice receiving i.v. injection of luc-4T1 cells were used as controls. **e** Outline of re-challenge study. **f** Representative bioluminescence images of mice at different times post injection of luc-4T1 cells. **g** Quantification of luminescence intensity of mice at different times post injection of luc-4T1 cells. Statistical analysis was performed using two-tailed *t*-tests. **h** Kaplan–Meier plots for all groups. Statistical analysis was performed using the log-rank (Mantel–Cox) test. For **g**, **h**, $n = 9$ biologically independent animals for the control group, and $n = 5$ for the gel vaccine group. All the numerical data are presented as mean ± SD. Source data are provided as a Source Data file.

potential in the 4T1 TNBC model. Further improvements may be achieved in the future via exploiting additional synergy among immunotherapies, chemotherapy, and radiation therapy[55,56]. The in situ vaccine may support clinical practice through several applications. First, in the case of breast-conserving surgery, the removal of all tumor cells in the breast tissues is difficult, and likely a cause for tumor recurrence[57]. Second, for patients who already have metastases in distant tissues at the time of surgery, even in the case of mastectomy, tumor recurrence at the surgical site is not rare[58]. Our in situ gel vaccine can take advantage of any remaining or recurring tumor cells to boost the immune responses against both primary and metastatic cancers. Third, our approach is applicable to tumors that cannot be surgically resected or can only be partially resected in the clinic.

To conclude, we describe a general strategy for in situ cancer vaccination that exploits both the immunogenic death of tumor cells and DC recruitment by biomaterial scaffolds. The strategy controls 4T1 tumor growth and reduces metastases with minimal systemic toxicity, potentially providing a potent therapeutic option for TNBCs. Checkpoint blockade therapies alone are not effective in the treatment of a variety of cancers including TNBCs, but the antitumor efficacy can be improved via combination with the in situ gel vaccine. The facile fabrication of the gel vaccine, which involves simple mixing of an alginate solution, pre-formed porogen beads, chemokines, drugs, adjuvants, and calcium ions over several minutes, is amenable to clinical translation. Components of the in situ gel vaccine used in this study, such as Dox-iRGD and CpG, can be readily replaced with other chemotherapeutic and immunotherapeutic drugs and adjuvants. This in situ gel vaccine can also be readily applied to various other types of cancers, and is likely to be especially relevant to cancers with limited availability of identified TAAs and neoantigens. This approach has the ability to develop robust, personalized in situ cancer vaccines without requiring identification of TAAs and personalized manufacturing.

## Methods

**Materials and instrumentation**. Doxorubicin was purchased from AstaTech, Inc. (Bristol, PA, USA). $N$-β-maleimidopropionic acid hydrazide was purchased from Thermo Fisher Scientific (Waltham, MA, USA). iRGD-SH and iRDG-SH were ordered from Peptide 2.0, Inc. (Chantilly, VA, USA). PRONOVA UP MVG sodium alginate (endotoxin-free) was purchased from Fmc Biopolymer AS (Sandvika, Norway). CpG ODN-SH was purchased from Integrated DNA Technologies (Coralville, IA, USA). Primary antibodies used in this study include the following: anti-PD-1 (RMP1-14, InVivoMAb anti-mouse PD-1, BioXCell), brilliant violet 421-conjugated anti-CD11b (Biolegend), fluorescein isothiocyanate (FITC)-conjugated anti-CD11c (Biolegend), PE/Cy7-conjugated anti-CD3 (Biolegend), PE-conjugated anti-F4/80 (Tonbo Biosciences), APC-conjugated anti-Gr1 (Biolegend), PE(phycoerythrin)/Cy7-conjugated anti-MHCII (Biolegend), PE-conjugated anti-CD86 (ebioscience), efluor450-conjugated anti-CD4 (ebioscience), FITC-conjugated anti-CD8 (Biolegend), APC-conjugated IFN-γ (ebioscience), PE-conjugated anti-TNF-α (ebioscience), pacific blue-conjugated anti-CD103 (ebioscience), PerCP/Cy5.5-conjugated anti-PD-L1 (Biolegend), APC-conjugated anti-CD8 (Biolegend), PE/Cy7-conjugated anti-CD11b (ebioscience), efluor450-conjugated anti-CD163 (Biolegend), APC-conjugated anti-CD206 (Biolegend), FITC-conjugated anti-CD86 (Biolegend), efluor450-conjugated anti-CD8 (ebioscience), FITC-conjugated anti-PD-1 (Biolegend), PE-conjugated anti-LAG3 (Biolegend), APC-conjugated anti-CTLA-4 (Biolegend), PE/Cy7-conjugated anti-TIM (Biolegend), FITC-conjugated anti-CD25 (Biolegend), PE-conjugated anti-Foxp3 (Biolegend), Alexa fluor 647-conjugated anti-calreticulin (Abcam), and FITC-conjugated anti-CD47 (Biolegend). Fixable viability dye efluor780 was obtained from Thermo Fisher Scientific. All antibodies were diluted according to the manufacturer's recommendations. Fluorescence-activated cell sorting (FACS) analyses were collected in BD FACS Diva on BD LSRII or BD LSR Fortessa flow cytometers and analyzed on FlowJo v7.6 and FCSExpress v6 and v7. Statistical testing was performed using GraphPad Prism v6 and v8. Fluorescence measurement of Dox was conducted on a plate reader. Small compounds were run on the Agilent 1290/6140 ultra high-performance liquid chromatography (HPLC)/mass spectrometer. H[1] NMR spectra were collected on the Agilent DD2 600. Matrix-assisted laser desorption/ionization mass spectra were collected on the Bruker Ultraflextreme MALDI-TOF/TOF Mass Spectrometer. Confocal images were taken using the Upright Zeiss LSM 710 microscope.

**Cell line and animals**. The 4T1 cell line was purchased from American Type Culture Collection (Manassas, VA, USA). 4T07 cell line was purchased from Karmanos Cancer Institute (Detroit, MI, USA). 4T1-eGFP-Puro cell line was purchased from Imanis Life Sciences (Rochester, MN, USA). EMT6 cell line was purchased from American Type Culture Collection (Manassas, VA, USA). EO771 cell line was a gift from Professor Kai Wucherpfennig's lab at Dana-Farber Cancer Institute. Cells were cultured in Dulbecco's modified Eagle's medium (DMEM) containing 10% fetal bovine serum, 100 units/mL Penicillin G, and 100 μg/mL streptomycin at 37 °C in 5% $CO_2$ humidified air. Female BALB/c and C57BL/6 mice were purchased from the Jackson Laboratory (Bar Harbor, ME, USA) and were 6–8 weeks old at the beginning of each experiment. Feed and water were available ad libitum. Artificial light was provided in a 12 h/12 h cycle. All procedures involving animals were done in compliance with National Institutes of Health and Institutional guidelines with approval of Harvard University's Institutional Animal Care and Use Committee.

**Synthesis of Dox-iRGD and Dox-iRDG**. Dox (0.1 mmol) and $N$-β-maleimidopropionic acid hydrazide (0.11 mmol) were dissolved in methanol, followed by the addition of acetic acid (20 μL). The mixture was stirred at 45 °C for 48 h. After removal of the solvent, the crude product was purified on preparative HPLC to yield Dox-Mal. Dox-Mal (0.02 mmol) and iRGD-SH (0.02 mmol) or iRDG-SH (0.02 mmol) were dissolved in PBS (pH 7.4) and stirred at room temperature for 48 h. After dialysis against deionized water with a 1 kDa cutoff membrane for 72 h, Dox-iRGD and Dox-iRDG were lyophilized and stored for use.

**Degradation of Dox-iRGD**. Dox-iRGD was dissolved in PBS with a pH value of 4 and 8, respectively, and added to dialysis tubes bearing membranes with a molecular weight cutoff of 1 kDa. The dialysis tubes were placed in PBS bath with a pH value of 4 and 8, respectively. At different time points, an aliquot of the release medium was taken out for fluorescence measurement on a plate reader. A concentration series of Dox were used for determining the standard curve.

**Preparation of Dox-iRGD-loaded pore-forming alginate gels**. Porogen beads were prepared following the reported method[18]. Three percent w/v alginate solution containing Dox-iRGD was mixed with GM-CSF-conjugated gold nanoparticles[18], resulting in a final concentration of 2% w/v unmodified alginate. This mixture, which constituted the bulk phase of the gels, was then mixed with pre-formed porogen beads. Finally, the bulk phase alginates were cross-linked by mixing with a sterile $CaSO_4$ slurry (0.2 g/mL). The volume of $CaSO_4$ cross-linking solution used was 4% v/v relative to the bulk alginate, and the volume fraction of porogens was 50% of the total gel volume. All mixing steps were performed using Luer-Lock syringes joined with Luer-Lock connectors. For in vitro studies, the gels were immediately cast between two silanized glass plates separated by 2 mm spacers. After allowing the gels to cross-link for 30 min, gel disks were punched out using a sterile biopsy punch. For in vivo studies, gels (100 μL) containing 3 μg of GM-CSF were freshly prepared and subcutaneously injected via an 18 G needle.

**Release of Dox-iRGD or Dox from pore-forming alginate gels**. Gel disks loaded with Dox-iRGD or free Dox were incubated in DMEM at 37 °C. At selected time points, aliquots of the release medium were taken out for fluorescence measurement on a plate reader, and the same amount of fresh medium was added. A concentration series of Dox-iRGD or Dox were used for determining the standard curve.

**In vitro immunogenic cell death assay**. 4T1 cells were incubated with different concentrations of Dox at 37 °C for 24 h. Cells without drug treatment were used as controls. Cells were washed with PBS for three times, collected, and stained with FITC-conjugated anti-CD47, Alex Fluor 647-conjugated anti-calreticulin, and fixable viability dye efluor780 for 20 min prior to FACS analyses.

**Tumor penetration study of Dox-iRGD**. 4T1 cells (0.6 million) in Hank's buffered salt solution (HBSS) buffer were subcutaneously injected into the right flank of Balb/c mice. After 5 days, pore-forming alginate gels loaded with GM-CSF (3 μg) + Dox-iRGD (50 μg in Dox equivalent) or GM-CSF + Dox-iRDG or GM-CSF + Dox were injected next to the tumors. After 48 h, gels and tumors were harvested and frozen with O.C.T. compound, and sectioned with a thickness of 8 μm. To determine the tumor penetration of drugs, tissue sections were stained with DAPI for 10 min and imaged under a confocal microscope. Tumor apoptosis was analyzed via TUNEL (terminal deoxynucleotidyl transferase dUTP nick end labeling) assay.

**Flow cytometry analysis of cells in gels**. Alginate gels were harvested from mice. For direct evaluation of cell viability, gels were disrupted using a syringe plunger and cells were filtered and stained with trypan blue. For cell analyses, gels were dissociated in 40 mM EDTA for 10 min on ice with vortex every 5 min, and filtered through a 40 μm cell strainer. Cells were pelleted and re-suspended in FACS buffer and blocked with anti-CD16/32 for 20 min. For gross evaluation of immune cell population, cells were stained with brilliant violet 421-conjugated anti-CD11b,

FITC-conjugated anti-CD11c, PE/Cy7-conjugated anti-CD3, PE-conjugated anti-F4/80, and APC-conjugated anti-Gr1 for 20 min. For DC analyses, cells were stained with brilliant violet 421-conjugated anti-CD11b, FITC-conjugated anti-CD11c, PE/Cy7-conjugated anti-MHCII, and PE-conjugated anti-CD86 for 20 min.

**Histological evaluation of tissues**. Lungs, liver, heart, kidneys, spleen, and spinal bone were collected from mice and fixed in formalin, paraffin-embedded, sectioned with a thickness of 4 μm, and stained with hematoxylin and eosin. Tissues were analyzed by a board-certified pathologist in a blinded manner. Images were taken under a microscope.

**In vitro restimulation of T cells from spleens and lymph nodes**. Lymph nodes and spleens were collected at 4 or 11 days post gel vaccination. Lymph nodes were treated with collagenase, disrupted, and filtrated through a 40-μm cell strainer to obtain the single cell suspension. Spleens were disrupted, filtrated through a 40 μm cell strainer, and treated with ACK lysis buffer to yield the single cell suspension. Cells isolated from lymph nodes and spleens were added to 4T1 cells that were pre-incubated in a 96-well plate overnight, and treated with GolgiPlug in T-cell medium at 37 °C for 4 h. Cells were then stained for flow cytometry analyses.

**Therapeutic efficacy study**. 4T1 tumors were established in Balb/c mice by subcutaneous injection of 4T1 cells (0.6 million cells in 50 μL of HBSS) into the right flank. After 5 or 6 days when the tumors reached a diameter of 6–7 mm, mice were randomly divided into four groups. Pore-forming alginate gels were freshly prepared and subcutaneously injected next to the tumor. The tumor volume and body weight of mice were measured every other day. The tumor volume was calculated using the formula (length) × (width)$^2$/2, where the long axis diameter was regarded as the length and the short axis diameter was regarded as the width. For Antitumor efficacy studies of gels containing Dox-iRGD, pore-forming alginate gels (100 μL) containing Dox-iRGD (50 μg in Dox equivalent) or Dox-iRDG (50 μg in Dox equivalent), or Dox (50 μg) were used ($N = 5$). At 18 days post gel injection, mice were killed, and gels and tumors were collected for analyses. For antitumor efficacy studies of gels containing Dox-iRGD and CpG, pore-forming alginate gels containing Dox-iRGD (100 μg) and CpG (50 μg), or Dox-iRGD (100 μg) alone or CpG (50 μg) alone were used ($N = 7$–8). For dose-increased antitumor efficacy studies, pore-forming alginate gels containing Dox-iRGD (200 μg) and CpG (100 μg) were used ($N = 7$). For the booster vaccine group, a second dose of gels containing Dox-iRGD (200 μg) and CpG (100 μg) were peritumorally injected at 7 days post the first gel injection. All gels were loaded with GM-CSF (3 μg). In some experiments, 4T1 cells (100 k cells in 50 μL of HBSS) or EMT6 cells (100 k cells in 50 μL of HBSS), or EO771 cells (750 k in 50 μL of HBSS) were injected into the mammary fat pad of mice, followed by peritumoral injection of gels. In another set of experiments, 4T07 cells (1.5 million cells in 50 μL of HBSS) were subcutaneously injected, followed by gel vaccination. EO771 tumors were inoculated in C57BL/6 mice while all the other models were established in Balb/c mice.

**Analyses of TME after vaccination**. 4T1 tumors were established in Balb/c mice by subcutaneous injection of 4T1 cells (0.6 million cells in 50 μL of HBSS) into the right flank. After 5 days, mice were randomly divided into different groups ($N = 6$). Pore-forming alginate gels containing Dox-iRGD (200 μg) and CpG (100 μg) or Dox-iRGD (200 μg) alone, or CpG (100 μg) alone were freshly prepared and subcutaneously injected next to the tumors. Mice without treatment were used as controls. At 11 days post gel injection, gels, tumors, lymph nodes, and spleens were harvested for analyses. Cells were isolated from gels following the abovementioned procedures and analyzed via flow cytometry. Tumors were incubated with collagenase and DNAse for 40 min at 37ºC, disrupted, and filtered through a 40 μm cell strainer. Cells were pelleted, blocked with anti-CD16/32 for 20 min, and then stained with fluorescently labeled antibodies for 20 min. For DC analysis, pacific blue-conjugated anti-CD103, FITC-conjugated anti-CD11c, PE-conjugated anti-CD86, PE/Cy7-conjugated anti-MHCII, PerCP/Cy5.5-conjugated anti-PD-L1, APC-conjugated anti-CD8, and fixable viability dye efluor780 were used. For macrophage analysis, PE/Cy7-conjugated anti-CD11b, PE-conjugated anti-F4/80, efluor450-conjugated anti-CD163, APC-conjugated anti-CD206, FITC-conjugated anti-CD86, and fixable viability dye efluor780 were used. For T-cell analyses, two antibody cocktails were used as follows: (1) efluor450-conjugated anti-CD8, FITC-conjugated anti-PD-1, PE-conjugated anti-LAG3, APC-conjugated anti-CTLA-4, PE/Cy7-conjugated anti-TIM, and fixable viability dye efluor780; (2) efluor450-conjugated anti-CD4, APC-conjugated anti-CD8, FITC-conjugated anti-CD25, and fixable viability dye efluor780. After staining with the second cocktail, cells were incubated in fixation/permeabilization working solution for 40 min at 4 °C. After washing with permeabilization buffer for twice, cells were stained with PE-conjugated anti-Foxp3 in permeabilization buffer at room temperature for 30 min. For tumor cell staining, Alexa Fluor 647-conjugated anti-calreticulin, FITC-conjugated anti-CD47, PerCP/Cy5.5-conjugated anti-PD-L1, and fixable viability dye efluor780 were used. Tumor extracts were analyzed for HMGB-1 using the enzyme-linked immunosorbent assay kit.

**Combination with anti-PD-1**. 4T1 tumors were established in Balb/c mice by subcutaneous injection of 4T1 cells (0.6 million cells in 50 μL of HBSS) into the right flank on Day 0. After 5 days when the tumors reached a diameter of 6–7 mm, mice were randomly divided into different groups ($N = 8$–9). Pore-forming alginate gels containing Dox-iRGD (200 μg) and CpG (100 μg) were freshly prepared and subcutaneously injected next to the tumor. Anti-PD-1 (100 μg) was intraperitoneally injected on Day 6, 9, 12, 15, and 18. The tumor volume and body weight of mice were measured every other day.

**Tumor resection model**. 4T1 tumors were established in Balb/c mice by subcutaneous injection of 4T1 cells or luciferase-expressing 4T1 cells (luc-4T1, 1 million cells in 50 μL of HBSS) into the right flank. After 9 days when the tumors reached a diameter of ~10–12 mm, tumors were surgically resected by an experienced surgeon. In brief, after anaesthetizing the animals, a scalpel was used to create a small incision at ~1 cm from the tumor site, followed by the resection of the vast majority of the tumor under a microscope. The visible tumor residues were further cleared using a scalpel under the microscope, without damaging the connective tissues. The incision was then closed using sutures, and before the last few stitches, pore-forming alginate gels containing GM-CSF (3 μg), Dox-iRGD (200 μg), and CpG (100 μg) or bolus vaccines (solution of GM-CSF, Dox-iRGD, and CpG) were injected to the surgical site through an 18 G needle. Tumor growth was then monitored by bioluminescence imaging or size measurement. At ~80 days post tumor resection, surviving mice were re-challenged with tail vein injection of 4T1 or luc-4T1 cells (100 k in 200 μL of HBSS). Development of metastases was monitored by bioluminescence imaging.

**Statistical analyses**. Statistical analysis was performed using GraphPad Prism v6 and v8, and Microsoft Excel. Sample variance was tested using the F test. For samples with equal variance, the significance between the groups was analyzed by a two-tailed Student's $t$-test. For samples with unequal variance, a two-tailed Welch's t-test was performed. For multiple comparisons, a one-way analysis of variance with post hoc Fisher's least significant difference test was used. The results were deemed significant at $0.01 < *P ≤ 0.05$, highly significant at $0.001 < **P ≤ 0.01$, and extremely significant at $***P ≤ 0.001$. Sample sizes of three to eight biologically independent samples per group and four to ten biologically independent animals per group were used for in vitro and in animal studies, respectively, as indicated in figure captions. Sample sizes for in vivo studies were determined empirically based on results from prior publications along with input and approval from Harvard University's Institutional Animal Care and Use Committee.

**Reporting summary**. Further information on research design is available in the Nature Research Reporting Summary linked to this article.

## Data availability
The data that support the findings of this study are available publicly (https://doi.org/10.7910/DVN/N3I4BQ). Source data are provided with this paper.

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

## Acknowledgements

We acknowledge funding from the National Institutes of Health (1 U54 CA244726, 1 R01 EB023287, 1 U01 CA214369, and 1 R01 CA223255). H.W. gratefully acknowledges funding support from the Wyss Technology Development Fellowship, and A.J.N. a Graduate Research Fellowship from the National Science Foundation.

## Author contributions

H.W., A.J.N., M.C.S., and D.J.M. designed the experiments. H.W., A.J.N., M.C.S., B.R.S., J.Y.L., A.W.L, C.S.V., and D.W. performed the experiments. H.W., A.J.N., M.C.S., and D.J.M. analyzed the data. H.W., A.J.N., M.C.S., and D.J.M. wrote the manuscript.

## Competing interests

D.J.M. declares the following competing interests: Novartis, sponsored research; Agnovos, consulting; Amgen, sponsored research; Samyang Corp., consulting; Decibel, sponsored research; Merck, sponsored research; Immulus, equity. All the other authors declare no competing interests.
