## [Peer Review File · Nature Communications]

Reviewers' comments:

Reviewer #1 (Remarks to the Author), expert in biomaterials/cancer/immunology:

Wang et al present a comprehensive manuscript detailing their approach to in situ DC vaccination using a peritumorally injected macroporous alginate gel. The gels were loaded with GM-CSF, CpG, and a DOX-iRGD conjugate, and showed the capability to recruit DCs, promoted the immunogenic death of tumor cells, and thus enhanced the antitumor T-cell response. The approach is based on their previously reported "infection-mimicking materials", but provides the new possibility to sample the whole tumor antigens in situ. The authors provide convincing data that their macroporous alginate gels in combination with anti-PD-1 antibody therapy significantly improved the efficacy against 4T1 murine triple negative breast cancer. Overall, the studies are quite thorough and well done and the data interpretation is reasonable. However, there are several questions that have to be clarified.

1. One major concern for the design of this strategy is the timing of DC maturation. Immature DCs typically exhibit high antigen uptake ability, while mature DCs are characterized by poor uptake of antigens and high expression of co-stimulator molecules such as MHCII, CD80, and CD8 (Nature 392, 245–252; Immunobiology 223(2018)25–31). Immature DCs were recruited to the alginate gels first and got matured there before they could migrate to tumor to capture the antigens (shown in Fig 2). It is a question that matured DCs in the gel can still efficiently capture the tumor antigens.
2. Evidence should be provided that DCs in the gel can indeed capture the tumor antigens from the tumor tissue or possibly in the gel directly. For example, GFP-expressing tumor cells could be used to inoculate the tumor and analyze the fluorescence in DCs. To further verify that the DCs indeed transported the tumor antigens to the draining lymph node, fluorescent antigens should be tracked in the draining lymph nodes as well.
3. DOX-iRGD was also loaded in the alginate gels with high concentration (50 μ g in Dox-equivalent in 100 μ L). Was the DOX-iRGD toxic to recruited DCs? The viability of the DCs in the gel should be verified.
4. Assuming soluble factors can exchange between the gel and tumor tissue, immunosuppressive soluble factors (e.g., IL-10, TGF β) that were in the tumor microenvironment could also accumulate in the gel. These immunosuppressive factors could potentially impair the DC function and antigen presentation, similarly as the situation in tumor-draining lymph nodes. Did the authors investigate the possible immunosuppression effect in the gel injected peri-tumor?
5. GM-CSF-conjugated gold nanoparticles were encapsulated in the gel to provide the chemotaxis signal to recruit the DCs. However, it is not clear how the recruited DCs can escape from the gels and migrate to tumor and then lymph node?
6. Fig 2h-I, it is not clear how the gate was determined. Positive and negative controls are needed to determine the gate for maturation markers (DCs treated with LPS or PBS buffer).
7. In fig 4c-e, CD8+ T cells in spleens and tumor-draining lymph nodes (tdLNs) were collected only 4 days post gel injection for FACS analysis for IFN γ secretion. Typically, it takes longer time for adaptive immune response to occur. Why did the authors choose such a short time period for analysis? Is there possibly enhanced effect given longer time period?
8. In fig 4c-e, "Strikingly, 24% of CD8+ T cells from tdLNs of mice treated with gels containing Dox-iRGD and CpG were IFN- γ + upon restimulation by 4T1 cells ex vivo, significantly higher than found from mice treated with gels containing Dox-iRGD alone (14%) or untreated mice (9%) (Fig. 4c-e)." The description in Results is not correlated with the data shown in the figure.
9. In Fig S10, why was the number of tumor infiltrating T cells in the group (Dox-iRGD + CpG) slightly less than the group of CpG alone?
10. In Fig 3-5, timelines of the studies can be added to clearly reveal the experimental design.
11. In situ dendritic cell vaccine with other approaches has been reported, such as, ACS Nano, Article ASAP (DOI: 10.1021/acsnano.8b08346), which should be cited in the references.

Reviewer #2 (Remarks to the Author), expert in immunotherapy:

In this original research report by Wang et al, a novel biomaterials scaffolding gel is described. This pore-forming, alginate gel possesses the ability to release active biomolecules over time in a sustained and uniform fashion and should be useful for the formulation of therapeutic compounds, the efficacy of which will be enhanced through sustained and local release. Though the reviewer's expertise is in tumor immunology and not biomaterials, these properties nonetheless appear to be novel. The authors are able to demonstrate enhanced efficacy of the gel material over and above that generated by the application of free biomolecules by means of a number of different tumor immunology experiments. It is these experiments that will be of paramount interest to the broadest number of readers and to which the bulk of the critiques will apply. While the combination of materials contained within the biomaterials gel (i.e. GM-CSF to attract myeloid APC, CpG to mature and polarize those APC, and tumor-penetrating doxorubicin to induce immunogenic cell death and provide antigens for presentation) is a novel combination of anti-tumor reagents, the anti-tumor results induced were pedestrian at best, and the immunologic characterizations suffered from a clear lack of expertise in immunology.

Major criticisms

1. First and foremost, the title of the paper is a misnomer. A vaccine can no more be antigen-free than an airplane can be wingless. The term "vaccine" is fundamentally defined by the presence of an antigen. Therefore, this treatment cannot be classified as a vaccine and the authors must pick different identifier. Suggest "immunoadjuvant" therapy.
2. Figure 2e shows a single, unquantified experiment. The authors should provide a graph that shows quantitation of multipole experimental repetitions.
3. Figures 2f - 2g - the ability to concentrate DC or other APC in the gel is irrelevant. If the DC/APC are not in the tumor or not in the peripheral lymphoid tissues, then they're not doing anything. Suggest eliminating these data and replacing with the data from Fig S5c-d. It is reasonable to define DC as CD11c+CD11b+MHCII+ with CD86 additionally used to define mature DC populations.
4. It is unclear what is being shown in figure 3d. Is it reduced cellularity? If so, this must be quantitated over multiple independent experiments.
5. Figure 5f/S8d - CD86 does NOT define M1 macrophages. M1/M2 polarization is defined functionally, by cytokine expression, and by other intracellular markers. There are no generally accepted combination of surface markers that distinguish M1 from M2.
6. In Fig 5b, there is no synergy observed between CpG and Dox-iRDG and therefore the authors cannot claim "CpG-induced amplification of endogenous immunogenic cancer cell death". No simplification of that observed by Dox-iRDG alone.
7. The HMGB1 data in figure 5c is problematic in that it could have multiple interpretations beyond what the authors claim. Since it doesn't add much to the manuscript, suggest removing.
8. The very modest level of synergy between anti-PD-1 therapy and alginate gel therapy was surprising. What anti-PD-1 antibody clone was used? The lack of synergy calls into questions the authors' hypothesis of a T-cell-mediated anti tumor mechanism, their other data notwithstanding.
9. The use of a luciferase-expressing cell line for the experiments outlined in figure 7 defeats the authors stated intent of developing a more clinically-relevant system. The presence of the highly immunogenic foreign luciferase antigen will generate a largely xenogeneic response. Suggest repeating these experiments with luc2-negative tumors or at least re-challenging the survivors

with luc2-negative tumors.

Minor criticisms

1. Line 57 ..."likely due to inadequate activation of adaptive immune responses". This is not universally or even partially true. Every individual treatment that doesn't work fails for its own specific reason, and this statement is not generalizable to all therapies that induce a subpar response. Remove.

2. Line 76 ..."likely due to inefficient use of released tumor antigens". This statement has no meaning. Modify with a coherent statement that postulates a valid immunologic hypothesis.

3. Line 165 ..."presumably...". Remove this entire clause. This speculative hypothesis was not shown.

4. Line 210 ..."presumably as part of the immune evasion mechanism". Tumors cells are self, and all self-cells respond to sustained and chronic immune attack by up regulation of both CD47 and PD-L1. Hence, this is really a generalized mechanism of peripheral tolerance and not a tumor-specific immune evasion mechanism. Hence these data are largely irrelevant.

5. Line 292 ..."this single-dose gel vaccine also outperforms the vast majority of previously reported immunotherapies in treating 4T-1 tumors". Disagree. The modest results reported here are equivalent or inferior to other treatment strategies of 4T-1 tumors reported in the literature. As an example of a robust results, see Kim et al, PNAS, 2014 or Wang et al, Nat Commun, 2018.

6. All flow cytometry plots should be presented in a uniform output format rather than various iterations of the three different output formats that appear throughout the manuscript. Pick one format and stick with it.

Reviewer #3 (Remarks to the Author), expert in cancer mouse models:

Overall this is an interesting study but premature at this stage. There were some fundamental issues with the study's experimental approach and interpretation that need to be addressed in order to be acceptable for publication.

1) The 4T1 were cells injected subQ instead of orthotopically. This is a problem for a number of reasons including this site of injection is less hospitable and is associated with increased cell death due to the hypoxic environment and delay in angiogenesis (compared to highly vascularized mammary tissues). Thus, the increase in DC recruitment is simply a reflection of the wound-like nature of injecting cells subQ along with the angiogenetic switch that this along with the surviving cancer cells induce. Need to repeat experiments orthotopically . In addition, need to repeat using established tumors of a minimum of 200 mm³ since this is a very different biological state than cancer cells injected and assayed later.

2) Need to include additional mammary tumor models and other cancer models as well. 4T1 line is not reflective of most TNBCs and using additional models will establish how generalizable the findings are.

3) The results, while statistically significant, are rather modest since the tumors still grew and metastasized.

4) It is unclear that the interpretation of this approach, that their gel vaccine is indeed acting like a vaccine (rather than just augmenting immune cell recruitment) is not well justified. In fact since tumor burden and not metastasis was affected by the gel vaccine regimen, this would suggest that recruiting DC cells to dying tumor cells is not the same as a vaccine since one would expect that intravasated cells would also be recognized by immune system.

5) The data do not support that the conclusion that the gel vaccine and anti PD-1 is synergistic- rather it appears to be additive.

Point-by-point reply to reviewer comments

(All responses were colored in blue, and all changes in the manuscript were highlighted in yellow)

Reviewers' comments:

Reviewer #1 (Remarks to the Author), expert in biomaterials/cancer/immunology:

Wang et al present a comprehensive manuscript detailing their approach to in situ DC vaccination using a peritumorally injected macroporous alginate gel. The gels were loaded with GM-CSF, CpG, and a DOX-iRGD conjugate, and showed the capability to recruit DCs, promoted the immunogenic death of tumor cells, and thus enhanced the antitumor T-cell response. The approach is based on their previously reported “infection-mimicking materials”, but provides the new possibility to sample the whole tumor antigens in situ. The authors provide convincing data that their macroporous alginate gels in combination with anti-PD-1 antibody therapy significantly improved the efficacy against 4T1 murine triple negative breast cancer. Overall, the studies are quite thorough and well done and the data interpretation is reasonable. However, there are several questions that have to be clarified.

Response: We appreciate the positive comments from the reviewer.

1. One major concern for the design of this strategy is the timing of DC maturation. Immature DCs typically exhibit high antigen uptake ability, while mature DCs are characterized by poor uptake of antigens and high expression of co-stimulator molecules such as MHCII, CD80, and CD8 (Nature 392, 245–252; Immunobiology223(2018)25–31). Immature DCs were recruited to the alginate gels first and got matured there before they could migrate to tumor to capture the antigens (shown in Fig 2). It is a question that matured DCs in the gel can still efficiently capture the tumor antigens.

Response: We thank the reviewer for this comment. The immature DCs would be exposed to antigens released from dying tumor cells as they migrate to the gel scaffold, where they are then activated. Regarding whether mature DCs can still efficiently capture tumor antigens, it has been reported that “although mature DCs markedly down-regulate their capacity for macropinocytosis, they continue to capture, process, and present antigens internalized via endocytic receptors” (Proc Natl Acad Sci 2010, 107, 4287–4292).

We have now added the following text in the revised manuscript: “The recruited DCs could sample antigens while they migrated towards and into the gel scaffold, where they were then activated.”

2. Evidence should be provided that DCs in the gel can indeed capture the tumor antigens from the tumor tissue or possibly in the gel directly. For example, GFP-expressing tumor cells could be used to inoculate the tumor and analyze the fluorescence in DCs. To further verify that the DCs indeed transported the tumor antigens to the draining lymph node, fluorescent antigens should be tracked in the draining lymph nodes as well.

Response: We thank the reviewer for this suggestion. We have now performed an additional study using GFP-expressing 4T1 cells to investigate the sampling of tumor antigens by recruited DCs in the presence of Dox-iRGD. As we can see from the updated Fig. S10 (also shown below), a significantly higher percentage of DCs in the gel vaccine were GFP-positive in comparison to control gels without Dox-iRGD. Further analyses of DCs in the tumor-draining lymph nodes also showed higher GFP fluorescence intensity in mice treated with the gel vaccine, in comparison to mice treated with control gels without incorporation of Dox-iRGD or untreated mice.

We have now added Fig. S10 and the following text to the revised manuscript: “To study whether the recruited DCs in gel scaffolds could sample tumor antigens, the gel vaccine was injected adjacent to subcutaneous GFP-expressing 4T1 tumors (Fig. S10a). At 4 days post injection, a significantly higher percentage of GFP-positive DCs was observed in the gel vaccine group compared to control gels without incorporation of Dox-iRGD (Fig. S10b-c). DCs in tdLNs also showed a stronger GFP signal in mice treated with the gel vaccine, in comparison to mice treated with control gels without incorporation of Dox-iRGD or untreated mice (Fig. S10d-f).”

Fig. S10. Recruited DCs in the gel vaccines can sample tumor antigens. (a) Time frame of animal study. The full gel vaccine was loaded with GM-CSF, Dox-iRGD, and CpG. Gels loaded with GM-CSF and CpG only were used as controls. (b) Representative flow cytometry profiles of cells isolated from the gel scaffolds. (c) Percentage of GFP⁺ DCs (CD11b⁺CD11c⁺) in the gel vaccine or control gel without incorporation of Dox-iRGD. (d) Representative GFP histograms of cells isolated from tumor-draining lymph nodes (tdLNs). (e) Percentage of GFP⁺ DCs in tdLNs. (f) Mean GFP fluorescence intensity of DCs in tdLNs.

3. DOX-iRGD was also loaded in the alginate gels with high concentration (50 µg in Dox-equivalent in 100 uL). Was the DOX-iRGD toxic to recruited DCs? The viability of the DCs in the gel should be verified.

Response: We thank the reviewer for this comment. Dox-iRGD was loaded in the bulk phase of alginate gels, while DCs were gradually recruited to pores that result from degradation of porogen beads after injection. The goal for this strategy was to ensure that, despite the high loading of Dox-iRGD in gels, the recruited DCs would not be exposed to a high concentration of Dox-iRGD; we apologize this aspect of our strategy was not clear. Analysis of cells in gel scaffolds at different times confirmed the high viability (>80%) of recruited cells.

We have now added the viability data as Fig. S5 (also shown below), and the following text in the revised manuscript: “It is noteworthy that cells in the gels that were harvested at different times showed high viability (>85%, Fig. S5)”.

Fig. S5. Viability of Cells in gels that were harvested at different times post peritumoral injection. Gels were loaded with GM-CSF, Dox-iRGD (200 µg), and CpG (100 µg).

4. Assuming soluble factors can exchange between the gel and tumor tissue, immunosuppressive soluble factors (e.g., IL-10, TGFβ) that were in the tumor microenvironment could also accumulate in the gel. These immunosuppressive factors could potentially impair the DC function and antigen presentation, similarly as the situation in tumor-draining lymph nodes. Did the authors investigate the possible immunosuppression effect in the gel injected peri-tumor?

Response: We thank the reviewer for this comment. We have now analyzed the immunosuppressive soluble factors IL-10 and TGF-β in gels at 4 days post gel injection, and these were present at a low baseline concentration of 250 pg/mL and 350 pg/mL, respectively. Although it was reported that IL-10 and TGF-β may prevent the maturation of DCs, the presence of CpG in the gel scaffolds can potentially overcome this effect. As a result, ~23% of DCs in the gel scaffolds

loaded with Dox-iRGD and CpG were CD86⁺MHCII⁺ at 4 days post gel injection, which was significantly higher than gels loaded with Dox-iRGD alone (Fig. S6c).

We have now added data for IL-10 and TGF- β levels in the gels as Fig. S6e (also shown below).

Fig. S6. (e) Concentration of IL-10 and TGF- β in gels loaded with GM-CSF, Dox-iRGD, and CpG at 4 days post subcutaneous injection.

5. GM-CSF-conjugated gold nanoparticles were encapsulated in the gel to provide the chemotaxis signal to recruit the DCs. However, it is not clear how the recruited DCs can escape from the gels and migrate to tumor and then lymph node?

Response: We thank the reviewer for this comment. In the absence of CpG, we previously observed that recruited DCs will leave the scaffold after the GM-CSF is mostly consumed. Further, CpG can activate recruited DCs, upregulate the receptor CCR7, and enhance their migration to the draining lymph nodes (Nat Mater. 2009, 8, 151–158).

We have now added the following text to the manuscript: “It is noteworthy that the recruited DCs tend to leave the gel scaffold when GM-CSF is mostly consumed, and CpG can further upregulate CCR7 receptors of DCs and facilitate their migration from the gel to lymph nodes.¹²”

6. Fig 2h-I, it is not clear how the gate was determined. Positive and negative controls are needed to determine the gate for maturation markers (DCs treated with LPS or PBS buffer).

Response: We have utilized isotype controls of primary antibodies as negative controls for gating, as are commonly used. Specifically for CD86⁺MHCII⁺ gating in Fig. 2h-i, cells isolated from the gels were stained with anti-CD86/anti-MHCII or isotype controls of these two antibodies, prior to flow cytometry analyses. We have now added the plot for isotype controls in Fig. 2h.

Fig. 2h. Representative flow cytometry plot of CD86⁺MHCI⁺ cells within CD11b⁺CD11c⁺ cells in Dox-iRGD loaded gels at 18 days post gel injection (upper panel). Lower panel contains the plot for isotype controls.

7. In fig 4c-e, CD8⁺ T cells in spleens and tumor-draining lymph nodes (tdLNs) were collected only 4 days post gel injection for FACS analysis for IFN γ secretion. Typically, it takes longer time for adaptive immune response to occur. Why did the authors choose such a short time period for analysis? Is there possibly enhanced effect given longer time period?

Response: We thank the reviewer for this comment. We have now analyzed tumor-specific T cell responses at 11 days post gel vaccination as well. As shown in the updated Fig. S8, gels loaded with Dox-iRGD and CpG showed significantly improved tumor-specific CD8⁺ and CD4⁺ T cell responses in both tumor-draining lymph nodes and spleen at 11 days post gel injection. We have now added Fig. S8 (also shown below) and revised the text accordingly: “Similarly, at 11 days post gel injection, a significant increase in the numbers of 4T1-specific CD8⁺ and CD4⁺ T cells in tdLNs and spleens was also detected in mice treated with gels containing Dox-iRGD and CpG, in comparison to mice treated with gels containing Dox-iRGD alone, mice treated with gels containing CpG alone, or untreated mice (Fig. S8)”.

Figure S8. Pore-forming gels containing GM-CSF, Dox-iRGD and CpG generate potent systemic tumor-specific CTL responses at 11 days post injection. (a) Time frame of animal study. GM-CSF was incorporated in all gels. (b) Percentage of IFN- γ ⁺ cells among CD8⁺ T cells in tdLNs at 11 days post injection of gels. (c) Percentage of IFN- γ ⁺ cells among CD4⁺ T cells in tdLNs at 11 days post injection of gels. (d) Percentage of IFN- γ ⁺ cells among CD8⁺ T cells in spleens at 11 days post injection of gels. (e) Percentage of IFN- γ ⁺ cells among CD4⁺ T cells in spleens at 11 days post injection of gels. All the numerical data are presented as mean \pm SD (0.01 < **P* \leq 0.05; ***P* \leq 0.01; ****P* \leq 0.001).

8. In fig 4c-e, “Strikingly, 24% of CD8⁺ T cells from tdLNs of mice treated with gels containing Dox-iRGD and CpG were IFN- γ ⁺ upon restimulation by 4T1 cells ex vivo, significantly higher than found from mice treated with gels containing Dox-iRGD alone (14%) or untreated mice (9%) (Fig. 4c-e).” The description in Results is not correlated with the data shown in the figure.

Response: We thank the reviewer for pointing out this mistake. These numbers have now been corrected.

9. In Fig S10, why was the number of tumor infiltrating T cells in the group (Dox-iRGD + CpG) slightly less than the group of CpG alone?

Response: There was no statistically significant difference between Dox-iRGD+CpG and CpG groups, indicating that CpG alone had a potent effect in facilitating tumoral infiltration of CD8⁺ T cells. This is consistent with previous findings that intratumoral injection of CpG could increase intratumoral infiltration of CD8⁺ T cells (J Immunother. 2011, 34, 279–288; Proc Natl Acad Sci. 2016, 113, 7240).

10. In Fig 3-5, timelines of the studies can be added to clearly reveal the experimental design.

Response: We appreciate this comment from the reviewer, and have now more clearly defined the timelines of each study in the figure captions.

11. In situ dendritic cell vaccine with other approaches has been reported, such as, ACS Nano, Article ASAP (DOI: 10.1021/acsnano.8b08346), which should be cited in the references.

Response: We thank the reviewer for this suggestion, and have added references accordingly.

Reviewer #2 (Remarks to the Author), expert in immunotherapy:

In this original research report by Wang et al, a novel biomaterials scaffolding gel is described. This pore-forming, alginate gel possesses the ability to release active biomolecules over time in a sustained and uniform fashion and should be useful for the formulation of therapeutic compounds, the efficacy of which will be enhanced through sustained and local release. Though the reviewer's expertise is in tumor immunology and not biomaterials, these properties nonetheless appear to be novel. The authors are able to demonstrate enhanced efficacy of the gel material over

and above that generated by the application of free biomolecules by means of a number of different tumor immunology experiments. It is these experiments that will be of paramount interest to the broadest number of readers and to which the bulk of the critiques will apply.

Response: We appreciate these positive comments from the reviewer.

While the combination of materials contained within the biomaterials gel (i.e. GM-CSF to attract myeloid APC, CpG to mature and polarize those APC, and tumor-penetrating doxorubicin to induce immunogenic cell death and provide antigens for presentation) is a novel combination of anti-tumor reagents, the anti-tumor results induced were pedestrian at best, and the immunologic characterizations suffered from a clear lack of expertise in immunology.

Response: We have to emphasize that 4T1 TNBC is a very aggressive and metastatic tumor model without any curative therapy to date. In efficacy studies, our in situ cancer vaccines could stop the growth of 4T1 tumors in a good fraction of animals, but the occurrence of metastases dimmed the overall survival benefit. Our efficacy results are stronger than or are comparable to the majority of therapies reported to date for 4T1 TNBC, with the novelty of not requiring the inclusion of an antigen in the therapeutic vaccine. We have also modified some of the immunological characterizations based on the reviewer's specific suggestions below.

Major criticisms

1. First and foremost, the title of the paper is a misnomer. A vaccine can no more be antigen-free than an airplane can be wingless. The term "vaccine" is fundamentally defined by the presence of an antigen. Therefore, this treatment cannot be classified as a vaccine and the authors must pick different identifier. Suggest "immunoadjuvant" therapy.

Response: We appreciate the reviewer's perspective on this topic. However, the USA Center for Disease Control (CDC) defines "vaccine" as "a product that stimulates a person's immune system to produce immunity to a specific disease, protecting the person from that disease. Vaccines are usually administered through needle injections, but can also be administered by mouth or sprayed into the nose." The World Health Organization (WHO) defines vaccine as follows: "A vaccine is a biological preparation that improves immunity to a particular disease. A vaccine typically contains an agent that resembles a disease-causing microorganism, and is often made from weakened or killed forms of the microbe, its toxins or one of its surface proteins. The agent stimulates the body's immune system to recognize the agent as foreign, destroy it, and "remember" it, so that the immune system can more easily recognize and destroy any of these microorganisms that it later encounters." Neither of these definitions requires a vaccine to contain an antigen, nor do many others from leading scientific organizations; while most vaccines do contain an antigen, that does not make our use of the term vaccine incorrect.

2. Figure 2e shows a single, unquantified experiment. The authors should provide a graph that shows quantitation of multipole experimental repetitions.

Response: Figure 2e was included to demonstrate apoptosis of tumor cells at the periphery of 4T1 tumors where Dox-iRGD was abundant. We have now included quantitative results for tumor apoptosis (Fig. S3c, also shown below) in the revised manuscript.

Fig. S3. (c) Percentage of apoptotic tumor cells at 4 days post gel injection, as determined by the TUNEL assay.

3. Figures 2f - 2g - the ability to concentrate DC or other APC in the gel is irrelevant. If the DC/APC are not in the tumor or not in the peripheral lymphoid tissues, then they're not doing anything. Suggest eliminating these data and replacing with the data from Fig S5c-d. It is reasonable to define DC as CD11c+CD11b+MHCII+ with CD86 additionally used to define mature DC populations.

Response: We apologize if this was not clear in the original manuscript, but the concentration of DCs in the gels is key to our approach. As Reviewer #1 indicated (Point 1), it is important for the DCs to sample antigen before being stimulated to mature; our design allows this by maturing the DCs in the gels after they have had an opportunity to sample antigens as they migrate towards and into the gel. As motivated by a further comment from Reviewer #1 (related to this same issue), we have now confirmed that DCs in the gels capture cancer antigens (see response to Reviewer #1, Point 2 above with new data). As a point of further clarification, the CD86⁺MHCII⁺ data in Fig. 2h-i were generated in the same studies as Fig. 2f-g; CpG was not incorporated into the gels in these studies. To keep the flow of data presentation consistent, we would like to keep Fig. 2f-g as is.

We have now added the following text in the revised manuscript: “The recruited DCs could sample antigens while they migrated towards and into the gel scaffold, where they were then activated.” We have also added Fig. S10 and the following text to the manuscript: “To study whether the recruited DCs in gel scaffolds could sample tumor antigens, the gel vaccine was injected adjacent to subcutaneous GFP-expressing 4T1 tumors (Fig. S10a). At 4 days post injection, a significantly higher percentage of GFP-positive DCs was observed in the gel vaccine group compared to control gels without incorporation of Dox-iRGD (Fig. S10b-c). DCs in tdLNs also showed a stronger GFP signal in mice treated with the gel vaccine, in comparison to mice treated with control gels without incorporation of Dox-iRGD or untreated mice (Fig. S10d-f).”

4. It is unclear what is being shown in figure 3d. Is it reduced cellularity? If so, this must be quantitated over multiple independent experiments.

Response: The data in Fig. 3d was intended to demonstrate the reduced cellularity in bone marrow that resulted from doxorubicin treatment (Dox group in this study); this has been used as a measure of acute toxicity of doxorubicin (Nat Chem Biol. 13, 4, 415; Int. J. Mol. Sci. 2018, 19, 484). The

standard approach to quantify this side-effect is the scoring of severity (Nat Chem Biol. 13, 4, 415), which has been added as Fig. S3d in the revised manuscript (also shown here).

Acute toxicity	Dox	Dox-iRGD	Dox-iRDG	Untreated
Bone marrow: Decreased cellularity	+ / +++ (5/5)	- (0/5)	- (0/5)	- (0/5)

Fig. S3. (d) Toxicity evaluation of gels loaded with Dox, Dox-iRGD, or Dox-iRDG in BALB/c mice. + (mild); +++ (marked); - (negative).

5. Figure 5f/S8d - CD86 does NOT define M1 macrophages. M1/M2 polarization is defined functionally, by cytokine expression, and by other intracellular markers. There are no generally accepted combination of surface markers that distinguish M1 from M2.

Response: We agree that there are no specific surface marker for M1/M2 macrophages, and their identification is a topic of considerable current interest. Along these lines, even cytokine expression cannot guarantee the differentiation between M1 and M2 macrophages. However, CD86, CD163, and CD206 gating within CD11b⁺F4/80⁺ cells has been widely used to evaluate M1/M2 polarization, as per the following papers:

Nature Communications 9, 559 (2018)
 Nature Communications 9, 873 (2018)
 Science Translational Medicine 2017, 9 (376), eaak9537
 PNAS 2018, 115 (18), E4236-E4244
 PNAS 2017 114 (26), E5077-E5084
 PLOS ONE 8(11): e80908

We have now added the following text and relevant references in the revised manuscript: “We also examined the population of M1/M2 macrophages in the tumor microenvironment. Although the definitive markers of M1 and M2 macrophages remain debatable, CD86, CD163, and CD206 gating within CD11b⁺F4/80⁺ cells has been widely used to evaluate M1/M2 polarization.^{39-42”}

6. In Fig 5b, there is no synergy observed between CpG and Dox-iRDG and therefore the authors cannot claim "CpG-induced amplification of endogenous immunogenic cancer cell death". No simplification of that observed by Dox-iRDG alone.

Response: We apologize as we had intended to state that CpG induced immunogenic death of 4T1 cells, as evidenced by a higher calreticulin signal in 4T1 tumors treated with CpG-loaded gels compared to untreated tumors. To avoid any confusion, we have deleted “CpG-induced amplification of endogenous immunogenic cancer cell death” in the revised manuscript.

7. The HMGB1 data in figure 5c is problematic in that it could have multiple interpretations beyond what the authors claim. Since it doesn't add much to the manuscript, suggest removing.

Response: We thank the reviewer for this comment. We have now removed this data and relevant text.

8. The very modest level of synergy between anti-PD-1 therapy and alginate gel therapy was surprising. What anti-PD-1 antibody clone was used? The lack of synergy calls into questions the authors' hypothesis of a T-cell-mediated anti tumor mechanism, their other data notwithstanding.

Response: The clone of anti-PD1 used in these studies is RMP1-14, from BioXCell (InVivoMAb anti-mouse PD-1). The main beneficial effect resulting from combination therapy was slower growth of primary tumors and significantly reduced metastases. Many tumors in the combination group actually stopped growing at the end of the study, with metastases being the main cause of death. Future additional optimization of gel vaccines, and optimization of the combination of gel vaccines and checkpoint blockades (e.g., timing and antibody choice) may potentially generate more significant synergy.

We have now added “Future additional optimization of the gel vaccine, and optimization of the combination of gel vaccines and checkpoint blockades (e.g., timing and antibody choice) may potentially generate more significant synergy” in the main text, and added “Anti-PD-1 (RMP1-14, InVivoMAb anti-mouse PD-1) was purchased from BioXCell (West Lebanon, NH, USA)” in the Methods section.

9. The use of a luciferase-expressing cell line for the experiments outlined in figure 7 defeats the authors stated intent of developing a more clinically-relevant system. The presence of the highly immunogenic foreign luciferase antigen will generate a largely xenogeneic response. Suggest repeating these experiments with luc2-negative tumors or at least re-challenging the survivors with luc2-negative tumors.

Response: We thank the reviewer for this comment. We previously used the luciferase-expressing 4T1 cell line as it allowed us to monitor the recurrence of 4T1 tumors post surgical resection via bioluminescence imaging, as used in many other papers (Science Translational Medicine 2014, 6, 242ra284-242ra284; Nature Communications 2017, 8, 14979; PNAS 2008, 105,4850-4855; Nature Cell Biology 2017, 19, 1260–1273; PLOS ONE 2010, 5, e9364). However, we agree with the reviewer that the luciferase itself may serve as an antigen. Following the reviewer’s suggestion, we have now repeated this study using 4T1 cells without luciferase expression. As shown in Fig. S17, a similarly beneficial effect of the gel vaccine in preventing tumor recurrence and following tumor re-challenge was observed.

We have now added Fig S17 and the following text to the revised manuscript: “To rule out the effect of potential immunogenicity of luciferase antigens, we repeated the tumor resection study using 4T1 cells. As shown in Fig. S17, the gel vaccine still resulted in significantly improved survival. Upon re-challenge with i.v. injected 4T1 cells, mice previously treated with the gel vaccine still showed 100% metastases-free survival (Fig. S17).”

Figure S17. In situ gel vaccines injected at the tumor resection site prevent the formation of 4T1 metastatic cancers. (a) Time frame of the animal study. Following surgical resection of 4T1 tumors, gels containing GM-CSF, Dox-iRGD (200 μg) and CpG (100 μg) were injected at tumor base. Mice were re-challenge with i.v. injected 4T1 cells at 60 days post gel injection. (b) Kaplan-Meier plots for overall survival of untreated mice or mice treated with the gel vaccine at tumor resection site. (c) Kaplan-Meier plots for overall survival of control mice (no previous treatment nor tumor inoculation) and mice that survived from (b), after 4T1 tumor challenge.

Minor criticisms

1. Line 57 ..."likely due to inadequate activation of adaptive immune responses". This is not universally or even partially true. Every individual treatment that doesn't work fails for its own specific reason, and this statement is not generalizable to all therapies that induce a subpar response. Remove.

Response: We have now removed this sentence.

2. Line 76 ..."likely due to inefficient use of released tumor antigens". This statement has no meaning. Modify with a coherent statement that postulates a valid immunologic hypothesis.

Response: We have now removed this sentence.

3. Line 165 ..."presumably...". Remove this entire clause. This speculative hypothesis was not shown.

Response: We appreciate this comment from the reviewer, but we think it is appropriate to analyze potential reasons for the improved antitumor efficacy here, which were verified in the following paragraphs.

4. Line 210 ..."presumably as part of the immune evasion mechanism". Tumors cells are self, and all self-cells respond to sustained and chronic immune attack by up regulation of both CD47 and PD-L1. Hence, this is really a generalized mechanism of peripheral tolerance and not a tumor-specific immune evasion mechanism. Hence these data are largely irrelevant.

Response: We thank the reviewer for this comment. We agree with the reviewer that CD47 upregulation is not a tumor-specific event. However, in the context of tumors, CD47 upregulation together with calreticulin and HMGB-1 has been frequently used as evidences for immunogenic death of tumor cells (Sci. Transl. Med. 2010, 2, 63ra94-63ra94; Oncology Letters 2019, 18, 6269-6274; Cancer Sci. 2019, 110, 256-268; Int. J. Mol. Sci. 2019, 20, 959; Breast Cancer Res Treat. 2018, 172, 69–82). To avoid confusion, we have now rephrased the text to "presumably as part of the immune evasion mechanism of tumor cells in response to the boosted immune responses".

5. Line 292 ..."this single-dose gel vaccine also outperforms the vast majority of previously reported immunotherapies in treating 4T-1 tumors". Disagree. The modest results reported here are equivalent or inferior to other treatment strategies of 4T-1 tumors reported in the literature. As an example of a robust results, see Kim et al, PNAS, 2014 or Wang et al, Nat Commun, 2018.

Response: We appreciate this comment from the reviewer. In Wang et al. Nature Commun 2018, the authors reported that a hydrogel loaded with killed tumor cells encapsulating a BRD4 inhibitor and a photoabsorbent, in combination with NIR irradiation, can prevent the post-surgical recurrence of 4T1 tumors and generate memory T cell responses, similar to what we showed in Figure 7. It is actually surprising that the PBS group (tumor resection only) and other control groups could survive as long as 60 days in Wang's study, while untreated animals in our study developed metastases and none survived longer than 40 days after surgical resection. Despite these variations, our data showed comparably potent efficacy to Wang's data for post-surgical prevention of 4T1 recurrence and generation of memory T cell responses.

Kim et al. (PNAS 2014, 111 (32), 11774-11779) reported that anti-PD1 and anti-CTLA4 together could regress tumors in 3 out of 10 animals, which is impressive. However, it is quite surprising that the animals in untreated group showed a median survival of ~40 days with subcutaneous injection of 5 million 4T1 cells, in comparison to ~25 days with subcutaneous injection of 500-600 k 4T1 cells in our study. Henau et al. (Nature 539, 443–447, 2016) reported a similar 4T1 tumor growth curve and animal survival as what we observed (500 k 4T1 cells, ~30 day median survival), but showed that combined anti-PD1 and anti-CTLA-4 failed to regress 4T1 tumors and only showed marginal survival improvement. These reports together with our study indicated that the antitumor efficacy of therapies against 4T1 tumors might be highly dependent on the proliferative status and metastatic potential of injected 4T1 cells.

In view of these, we are confident that our gel vaccine outperforms or is comparable to most successful immunotherapies in treating 4T1 tumors to date.

We have now modified the text to “this single-dose gel vaccine outperforms or is comparative to the vast majority of previously reported immunotherapies in treating 4T-1 tumors, including checkpoint blockade therapies, DC vaccines, and immunostimulatory agents⁴⁵⁻⁴⁹” and added new references 47-49.

6. All flow cytometry plots should be presented in a uniform output format rather than various iterations of the three different output formats that appear throughout the manuscript. Pick one format and stick with it.

Response: We appreciate this comment from the reviewer. We have now used pseudocolor output for all flow cytometry plots.

Reviewer #3 (Remarks to the Author), expert in cancer mouse models:

Overall this is an interesting study but premature at this stage. There were some fundamental issues with the study’s experimental approach and interpretation that need to be addressed in order to be acceptable for publication.

1) The 4T1 were cells injected subQ instead of orthotopically. This is a problem for a number of reasons including this site of injection is less hospitable and is associated with increased cell death due to the hypoxic environment and delay in angiogenesis (compared to highly vascularized mammary tissues). Thus, the increase in DC recruitment is simply a reflection of the wound-like nature of injecting cells subQ along with the angiogenetic switch that this along with the surviving cancer cells induce. Need to repeat experiments orthotopically. In addition, need to repeat using established tumors of a minimum of 200 mm³ since this is a very different biological state than cancer cells injected and assayed later.

Response: We thank the reviewer for this comment. We have now tested the in situ gel vaccine in an orthotopic model in large tumors, i.e., 4T1 cells were injected into the mammary fat pad of animals. As shown in Fig. S18, the gel vaccine was able to significantly slow the growth of 4T1 tumors and improve survival.

We have now added Fig. S18 and the following text in the revised manuscript: “To further evaluate the antitumor efficacy of in situ gel vaccines in an orthotopic tumor model, we injected 4T1 cells into the mammary fat pad, and peritumorally injected the gel vaccines when the tumors grew to ~150 mm³. Similar to previous observations, the in situ gel vaccine was able to significantly slower the tumor growth and improve the survival of mice (Fig. S18).”

Figure S18. In situ gel vaccine slows tumor growth and prolongs animal survival in an orthotopic 4T1 tumor model. (a) Time frame of the animal study. Gels were loaded with GM-CSF, Dox-iRGD (200 μ g) and CpG (100 μ g). (b) Tumor growth profiles for each group. (c) Kaplan-Meier plots for overall survival of all groups (n=9-10).

2) Need to include other cancer models as well. 4T1 line is not reflective of most TNBCs and using additional models will establish how generalizable the findings are.

Response: We thank the reviewer for this suggestion. We have now supplemented an efficacy study in the 4T07 TNBC model. As shown in Fig. S19, the in situ gel vaccine could significantly slow the growth of 4T07 tumors.

We have now added Fig. S19 and the following text to the revised manuscript: “We also studied whether the in situ gel vaccine would be effective against other TNBC models such as 4T07 tumors. As expected, the gel vaccine showed potent efficacy against 4T07 tumors (Fig. S19). These experiments further validated the potent antitumor efficacy of the in situ gel vaccine against TNBCs.”

Figure S19. In situ gel vaccine slows potent efficacy against 4T07 tumors. (a) Time frame of the animal study. Gels were loaded with GM-CSF, Dox-iRGD (200 μ g) and CpG (100 μ g). (b) Tumor growth profiles for each group (n =10).

3) The results, while statistically significant, are rather modest since the tumors still grew and metastasized.

Response: We have to emphasize that 4T1 TNBC is a very aggressive and metastatic tumor model without any curative therapy to date. Untreated animals had a median survival of ~27 days and all reached endpoint at ~32 days, with subcutaneous inoculation of 500 k cells. In efficacy studies, our in situ cancer vaccines could stop the growth of 4T1 tumors in most animals, but the occurrence of metastases dimmed the overall survival benefit. Nevertheless, our efficacy results are stronger than or are comparable to the majority of therapies reported to date for 4T1 TNBC, with the novelty of not requiring the inclusion of an antigen in the therapeutic vaccine. As a reference, most chemotherapies showed negligible therapeutic benefits to date, as well as some emerging immunotherapies. For example, Henau et al. (Nature 539, 443–447, 2016) reported that combined anti-PD1 and anti-CTLA-4 only showed marginal survival improvement. We have now added more references to support our claims in the revised manuscript.

4) It is unclear that the interpretation of this approach, that their gel vaccine is indeed acting like a vaccine (rather than just augmenting immune cell recruitment) is not well justified. In fact since tumor burden and not metastasis was affected by the gel vaccine regimen, this would suggest that recruiting DC cells to dying tumor cells is not the same a vaccine since one would expect that intravasated cells would also be recognized by immune system.

Response: We believe that our data support that the gel vaccine is indeed acting as a vaccine, and apologize if that was not clear in the original manuscript. As Reviewer #1 indicated (Point 1), it is

important for the DCs to sample antigen before being stimulated to mature; our design allows this by maturing the DCs in the gels after they have had an opportunity to sample antigens as they migrate towards and into the gel. As motivated by a further comment from Reviewer #1 (related to this same issue), we have now confirmed that DCs in the gels capture cancer antigens (see response to Reviewer #1, Point 2 above with new data). Regarding the antitumor efficacy, the gel vaccine actually reduced the number of metastases compared to the untreated group, but was not potent enough to fully suppress the formation of metastases.

To clarify this point, we have now added the following text in the revised manuscript: “The recruited DCs could sample antigens while they migrated towards and into the gel scaffold, where they were then activated.” We have also added Fig. S10 and the following text to the manuscript: “To study whether the recruited DCs in gel scaffolds could sample tumor antigens, the gel vaccine was injected adjacent to subcutaneous GFP-expressing 4T1 tumors (Fig. S10a). At 4 days post injection, a significantly higher percentage of GFP-positive DCs was observed in the gel vaccine group compared to control gels without incorporation of Dox-iRGD (Fig. S10b-c). DCs in tdLNs also showed a stronger GFP signal in mice treated with the gel vaccine, in comparison to mice treated with control gels without incorporation of Dox-iRGD or untreated mice (Fig. S10d-f).”

5) The data do not support that the conclusion that the gel vaccine and anti PD-1 is synergistic- rather it appears to be additive.

Response: We appreciate this comment from the reviewer. The slower tumor growth and reduced metastases showed certain synergy between anti-PD-1 and the in situ gel vaccine. However, we agree that the survival data did not show a synergistic effect, and we have now modified the text and figure captions accordingly in the revised manuscript.

Reviewers' comments:

Reviewer #1 (Remarks to the Author):

The authors have performed substantial new work, and added the resulting data to the manuscript to appropriately address the earlier concerns. The manuscript has been significantly improved. In particular, the mechanism study of tumor antigen sampling and the addition of another tumor model strengthened the conclusion. Only one more minor comment:

1. I also agree with Reviewer 2 that the title of the manuscript could be misleading. In any case, antigen is always there. It is just a matter that the antigen is included as a component in the treatment, or in situ in the disease (here is tumor). Otherwise, what is the immune response against? I would suggest the authors to use some other words, such as "in situ antigen", to replace "antigen-free" and better reflect the design.

Reviewer #2 (Remarks to the Author):

The authors have been partially responsive to the previous reviewer comments; however, some significant concerns remain.

Major criticism #1.

The authors contend it is appropriate to term their treatment "an antigen-free vaccine" despite the reviewer's previous suggestions to alter this terminology. There appears to be a misunderstanding on the part of the authors that their proposed treatment regimen is unique and/or novel and without previous precedent and therefore terminology may be left to the author's discretion. Unfortunately, this is not the case. As a small example, please consider the below SIX patent applications that propose applying a combination of immunomodulatory and cytotoxic agents as a means to treat tumors. In none of these previous circumstances did the inventors (who are all experts in the field) choose to describe their invention as "an antigen-free vaccine" in either the patent literature or the scientific literature that underpinned each of the inventions, despite the fact that these regimens function identically to that described by Wang et al.

1. DRUG DELIVERY COMPOSITIONS AND USES THEREOF

Application Number: CA3033542A

Publication Number: CA3033542A1

Priority: US201662381456P

(Filed 08/03/2016)

This application describes a drug delivery composition which can include a TLR agonist, STING agonist, resiquimod, galunisertib, and a RIG-I like receptor agonist, among other agents

2. COMPOSITIONS AND METHODS FOR CANCER TREATMENT

Application Number: PCT/US2019/023157

Publication Number: WO2019183216A1

Priority: US201862645613P

(Filed 03/20/2018)

This application describes a drug delivery composition which can include multiple TLR agonists and a STING agonist

3. IMMUNOSTIMULATORY COMBINATIONS FOR VACCINE ADJUVANTS

Application Number: PCT/US2007/000616

Publication Number: WO2007120368A2

Priority: US75731406P

(Filed 01/09/2006)

This application describes a drug delivery composition which can include multiple TLR agonists and multiple RLH type agonists

4. ENHANCING TREATMENT OF CANCER AND HIF-1 MEDIATED DISORDERS WITH ADENOSINE A3 RECEPTOR ANTAGONISTS

Application Number: PCT/US2005/042551

Publication Number: WO2007040565A3

Priority: US63055704P

(Filed 11/22/2004)

Status: Application Discontinuation

This application describes a drug delivery composition which can include an A3 receptor antagonist, a chemotherapeutic agent, an antiangiogenic agent, a cytotoxic agent and a cancer therapeutic antibody.

5. USE OF RESIQUIMOD FOR THE TREATMENT OF CUTANEOUS METASTASES

Application Number: PCT/US2005/047467

Publication Number: WO2006071997A2

Priority: US64049104P

(Filed 12/30//2004)

Status: Application Filing

Active US patent US8461174B2 expires 08/30/2028

This application describes a drug delivery composition which can include a TLR7/8 agonist delivered in combination with an imidazoquinoline amine.

6. USE OF STING AGONIST AS A CANCER TREATMENT

Application Number: PCT/US2014/066436

Publication Number: WO2015077354A1

Priority: US201361906330P

(Filed 11/19/2013)

Status: Application Filing

This application describes a drug delivery composition which can include a STING agonist delivered in combination with radiotherapy, chemotherapy, and/or other toxin therapy, among other agents.

Further, this list should not be considered exhaustive. There are literally hundreds of similar such patent applications and primary research literature. These six just happened to be sitting at the top of the reviewer's email inbox. In no patent application nor research literature that the reviewer has previously read has an immunomodulatory strategy been described as "an antigen-free vaccine" given that, as previously discussed, this terminology doesn't make sense immunologically. While the reviewer very much understands the authors' desire to market their work in a provocative manner, such provocation cannot be misleading and inaccurate. Therefore, the copy on the WHO and CDC websites written for a lay audience notwithstanding, the reviewer continues to insist that the authors change the description of their technology to something that accurately reflects bona fide immunologic mechanism.

Major criticism #2

Acceptable resolution. Thank you.

Major criticism #3

The reviewer appreciates the addition of Fig S10d-f which addresses his original point. Thank you.

Major criticism #4

Acceptable resolution. Thank you.

Major criticism #5

Use of these marker combinations with appropriate references is acceptable. Thank you.

Major criticisms #6 - 10

Acceptable resolutions. Thank you.

Minor criticisms #1-4

Acceptable resolutions. Thank you.

Minor criticism #5

This is really a major criticism. The reviewer has noted that the author's results are unimpressive when compared to other work in the field. Based on a review of untreated controls, the authors have countered that their 4T-1 cells are more aggressive and more potent than those used by other investigators, and therefore that their work should be given greater consideration. The authors do have a point here. Clonal drift of cell lines over time and between laboratories and institutions is an issue that often confounds the interpretation of experimental results. It is for this reason that most investigators are made to use a variety of different model systems, typically orthotopic, in order to claim a meaningful advance. Therefore, the reviewer would like to see the author's 4T-1 results repeated in at least two additional orthotopic systems. Obviously it is not necessary that the authors repeat every experiment in the two additional systems. An efficacy experiment with appropriate analysis will be sufficient for each additional model system employed.

Minor criticism #6

Acceptable resolution. Thank you.

Reviewer #4 (Remarks to the Author):

In general, I believe the authors answered Reviewer #3's criticisms, albeit not as strongly as they could have.

They do employ now a second model which shows greater activity, but as a scientist, I find it odd that the investigators chose to use another 4T1 derivative (4T07), rather than a different murine breast tumor model altogether. This doesn't really demonstrate rigor of the science, rather what can one do quickly to satisfy a reviewer. The model, and a sub derivative, does not make it generalizable. It likely requires the tumor itself to have significant sensitivity to adriamycin, and may not work in many models, though that isn't tested here.

I find it odd that this was already reviewed and critiqued without any explanation in the manuscript about what iRDG peptide - defined as 'tumor penetrating' - or iRGD (the 'scramble control') actually is. Not understanding this aspect, I cannot confidently say I understand the technology or strategy as it is presented, outside of this being a controlled-dox release implant, with tumor activity, rather than a vaccine as the authors describe.

Regarding the nomenclature as a vaccine: Indeed, in the authors own explanation:

We appreciate the reviewer's perspective on this topic. However, the USA Center for Disease Control (CDC) defines "vaccine" as "a product that stimulates a person's immune system to produce immunity to a specific disease, protecting the person from that disease. Vaccines are usually administered through needle injections, but can also be administered by mouth or sprayed into the nose."

- this is clearly for a lay audience, not scientific explanation.

The World Health Organization (WHO) defines vaccine as follows: "A vaccine is a biological preparation that improves immunity to a particular disease. A vaccine typically contains an agent

that resembles a disease-causing microorganism, and is often made from weakened or killed forms of the microbe, its toxins or one of its surface proteins. The agent stimulates the body's immune system to recognize the agent as foreign, destroy it, and "remember" it, so that the immune system can more easily recognize and destroy any of these microorganisms that it later encounters."

- This clearly is an explanation of antigen(s). The technology proposed here utilizes doxorubicin to kill tumor cells, in situ, releasing antigen. So this is no more of a vaccine than administering systemic doxorubicin. I am actually quite surprised the authors chose to argue this, particularly from this direction. This aspect makes me concerned.

There are other serious misnomers/instances where the data do not support conclusions, for example in figure title claims. For instance, one egregious example:
Supplemental Figure S7: Pore-forming gels containing GM-CSF, Dox-iRGD and CpG generate potent systemic tumor-specific CTL responses.

Nothing in this figure shows anti-tumor immune responses, it simply shows interferon gamma production by T cells, and does not demonstrate that they are anti-tumor T cells. The same problem exists with the Figure S8 title.

Finally, I could not find anywhere where the same experimental design was conducted in, for instance a RAG deficient or nude balb/c mouse to demonstrate that the gel 'vaccine' was actually slowing tumor growth due to immunological effects.

The immunologic memory experiments in Figure 7, I truly cannot begin to understand what the authors' hypothesis is for why this would be generating immunologic memory when implanted locally, after tumors had been resected, if it is functioning as a 'vaccine'. More likely, the tumor is probably incompletely removed - i.e. residual tumor cells as stated by the authors - due to the fact that they are not mouse surgeons. How that hypothesis makes the model more clinically relevant is difficult to understand, since all positive margin resections are repeat surgery clinically; this is a very infrequent occasion. Overall, what is going on with systemic immunity is not clear.

Point-by-point reply to reviewers comments

(All responses were colored in blue, and all changes in the manuscript were highlighted in yellow)

Reviewer #1 (Remarks to the Author):

The authors have performed substantial new work, and added the resulting data to the manuscript to appropriately address the earlier concerns. The manuscript has been significantly improved. In particular, the mechanism study of tumor antigen sampling and the addition of another tumor model strengthened the conclusion. Only one more minor comment:

1. I also agree with Reviewer 2 that the title of the manuscript could be misleading. In any case, antigen is always there. It is just a matter that the antigen is included as a component in the treatment, or in situ in the disease (here is tumor). Otherwise, what is the immune response against? I would suggest the authors to use some other words, such as “in situ antigen”, to replace “antigen-free” and better reflect the design.

Response: We thank the Reviewer for the positive comments. We have now changed the title to “*Biomaterial-Based In Situ Cancer Vaccine to Treat Poorly Immunogenic Tumors*”.

Reviewer #2 (Remarks to the Author):

The authors have been partially responsive to the previous reviewer comments; however, some significant concerns remain.

Major criticism #1.

The authors contend it is appropriate to term their treatment "an antigen-free vaccine" despite the reviewer's previous suggestions to alter this terminology. There appears to be a misunderstanding on the part of the authors that their proposed treatment regimen is unique and/or novel and without previous precedent and therefore terminology may be left to the author's discretion. Unfortunately, this is not the case. As a small example, please consider the below SIX patent applications that propose applying a combination of immunomodulatory and cytotoxic agents as a means to treat tumors. In none of these previous circumstances did the inventors (who are all experts in the field) choose to describe their invention as "an antigen-free vaccine" in either the patent literature or the scientific literature that underpinned each of the inventions, despite the fact that these regimens function identically to that described by Wang et al.

(1) DRUG DELIVERY COMPOSITIONS AND USES THEREOF

Application Number: CA3033542A

Publication Number: CA3033542A1

Priority: US201662381456P

(Filed 08/03/2016)

This application describes a drug delivery composition which can include a TLR agonist, STING agonist, resiquimod, galunisertib, and a RIG-I like receptor agonist, among other agents

(2) COMPOSITIONS AND METHODS FOR CANCER TREATMENT

Application Number: PCT/US2019/023157

Publication Number: WO2019183216A1

Priority: US201862645613P

(Filed 03/20/2018)

This application describes a drug delivery composition which can include multiple TLR agonists and a STING agonist

(3) IMMUNOSTIMULATORY COMBINATIONS FOR VACCINE ADJUVANTS

Application Number: PCT/US2007/000616

Publication Number: WO2007120368A2

Priority: US75731406P

(Filed 01/09/2006)

This application describes a drug delivery composition which can include multiple TLR agonists and multiple RLH type agonists

(4) ENHANCING TREATMENT OF CANCER AND HIF-1 MEDIATED DISORDERS WITH ADENOSINE A3 RECEPTOR ANTAGONISTS

Application Number: PCT/US2005/042551

Publication Number: WO2007040565A3

Priority: US63055704P

(Filed 11/22/2004)

Status: Application Discontinuation

This application describes a drug delivery composition which can include an A3 receptor antagonist, a chemotherapeutic agent, an antiangiogenic agent, a cytotoxic agent and a cancer therapeutic antibody.

(5) USE OF RESIQUIMOD FOR THE TREATMENT OF CUTANEOUS METASTASES

Application Number: PCT/US2005/047467

Publication Number: WO2006071997A2

Priority: US64049104P

(Filed 12/30/2004)

Status: Application Filing

Active US patent US8461174B2 expires 08/30/2028

This application describes a drug delivery composition which can include a TLR7/8 agonist delivered in combination with an imidazoquinoline amine.

(6) USE OF STING AGONIST AS A CANCER TREATMENT

Application Number: PCT/US2014/066436

Publication Number: WO2015077354A1

Priority: US201361906330P

(Filed 11/19/2013)

Status: Application Filing

This application describes a drug delivery composition which can include a STING agonist delivered in combination with radiotherapy, chemotherapy, and/or other toxin therapy, among other agents.

Further, this list should not be considered exhaustive. There are literally hundreds of similar such patent applications and primary research literature. These six just happened to be sitting at the top of the reviewer's email inbox. In no patent application nor research literature that the reviewer has previously read has an immunomodulatory strategy been described as "an antigen-free vaccine" given that, as previously discussed, this terminology doesn't make sense immunologically. While the reviewer very much understands the authors' desire to market their work in a provocative manner, such provocation cannot be misleading and inaccurate. Therefore, the copy on the WHO and CDC websites written for a lay audience notwithstanding, the reviewer continues to insist that the authors change the description of their technology to something that accurately reflects bona fide immunologic mechanism.

Response: We thank the Reviewer for these comments. We have now changed the title to 'Biomaterial-Based In Situ Cancer Vaccine to Treat Poorly Immunogenic Tumors', and have also avoided the use of 'antigen free' throughout the manuscript.

Major criticism #2

Acceptable resolution. Thank you.

Major criticism #3

The reviewer appreciates the addition of Fig S10d-f which addresses his original point. Thank you.

Major criticism #4

Acceptable resolution. Thank you.

Major criticism #5

Use of these marker combinations with appropriate references is acceptable. Thank you.

Major criticisms #6 - 10

Acceptable resolutions. Thank you.

Minor criticisms #1-4

Acceptable resolutions. Thank you.

Minor criticism #5

This is really a major criticism. The reviewer has noted that the author's results are unimpressive when compared to other work in the field. Based on a review of untreated controls, the authors have countered that their 4T-1 cells are more aggressive and more potent than those used by other investigators, and therefore that their work should be given greater consideration. The authors do have a point here. Clonal drift of cell lines over time and between laboratories and institutions is an issue that often confounds the interpretation of experimental results. It is for this reason that most investigators are made to use a variety of different model systems, typically orthotopic, in order to claim a meaningful advance. Therefore, the reviewer would like to see the author's 4T-1 results repeated in at least two additional orthotopic systems. Obviously it is not necessary that the authors repeat every experiment in the two additional systems. An efficacy experiment with

appropriate analysis will be sufficient for each additional model system employed.

Response: We thank the Reviewer for these comments. We have now supplemented efficacy studies with two additional orthotopic models, EMT-6 and EO771 TNBCs. The orthotopic tumors were established via mammary fat pad injection of EMT-6 or EO771 cells. As shown in Fig. S20, the gel vaccine significantly slowed tumor growth, in comparison to the untreated group, and eradicated 30% of orthotopic EMT6 tumors. The in situ gel vaccine also significantly reduced the EO771 tumor growth rate and improved the survival of animals, as compared to the untreated group (Figure S21).

Figure S20. In situ gel vaccine slows tumor growth and prolongs animal survival in an orthotopic EMT6 tumor model. (a) Time frame of the animal study. Gels were loaded with GM-CSF, Dox-iRGD (200 μ g) and CpG (100 μ g). (b) Tumor growth profiles for each group. (c) Kaplan-Meier plots for overall survival of all groups.

Figure S21. In situ gel vaccine slows tumor growth and prolongs animal survival in an orthotopic EO771 tumor model. (a) Time frame of the animal study. Gels were loaded with GM-CSF, Dox-iRGD (200 μ g) and CpG (100 μ g). (b) Tumor growth profiles for each group. (c) Kaplan-Meier plots for overall survival of all groups.

In the manuscript, we also added the following text: “We also studied whether the in situ gel vaccine would be effective against other TNBC models. In an orthotopic EMT6 TNBC model, the gel vaccine also significantly reduced the tumor growth rate and improved the survival of mice compared to controls, with 30% of tumors being eradicated (Fig. S20). These results were mimicked in an orthotopic EO771 TNBC model, where the in situ vaccine again demonstrated therapeutic benefit (Fig. S21).”

Minor criticism #6

Acceptable resolution. Thank you.

Reviewer #4 (Remarks to the Author):

In general, I believe the authors answered Reviewer #3's criticisms, albeit not as strongly as they could have.

1. They do employ now a second model which shows greater activity, but as a scientist, I find it odd that the investigators chose to use another 4T1 derivative (4T07), rather than a different murine

breast tumor model altogether. This doesn't really demonstrate rigor of the science, rather what can one do quickly to satisfy a reviewer. The model, and a sub derivative, does not make it generalizable. It likely requires the tumor itself to have significant sensitivity to adriamycin, and may not work in many models, though that isn't tested here.

Response: Thank you for the suggestion, and we have now supplemented the efficacy study with two additional orthotopic TNBC models, EMT-6 and EO771. As shown in Figure S20 and Figure S21, the in situ gel vaccine significantly reduced the tumor growth rate and improved the survival of animals bearing either EMT-6 or EO771 tumors, in comparison to the untreated group.

Figure S20. In situ gel vaccine slows tumor growth and prolongs animal survival in an orthotopic EMT6 tumor model. (a) Time frame of the animal study. Gels were loaded with GM-CSF, Dox-iRGD (200 μg) and CpG (100 μg). (b) Tumor growth profiles for each group. (c) Kaplan-Meier plots for overall survival of all groups.

Figure S21. In situ gel vaccine slows tumor growth and prolongs animal survival in an orthotopic EO771 tumor model. (a) Time frame of the animal study. Gels were loaded with GM-CSF, Dox-iRGD (200 μ g) and CpG (100 μ g). (b) Tumor growth profiles for each group. (c) Kaplan-Meier plots for overall survival of all groups.

We have now added the following text in the manuscript: “We also studied whether the in situ gel vaccine would be effective against other TNBC models. In an orthotopic EMT6 TNBC model, the gel vaccine significantly reduced the tumor growth rate and improved the survival of mice, with 30% of tumors being eradicated (Fig. S20). Similarly, the in situ gel vaccine also demonstrated therapeutic benefit in an orthotopic EO771 TNBC model (Fig. S21).”

2. I find it odd that this was already reviewed and critiqued without any explanation in the manuscript about what iRDG peptide - defined as 'tumor penetrating' - or iRGD (the 'scramble control') actually is. Not understanding this aspect, I cannot confidently say I understand the technology or strategy as it is presented, outside of this being a controlled-dox release implant, with tumor activity, rather than a vaccine as the authors describe.

Response: We appreciate the Reviewer’s comment regarding iRGD. iRGD (CRGDKGPDC) is a tumor-penetrating peptide that has been widely explored for its ability to aid in the tumor penetration of molecules to which it is attached (e.g., Teesalu et al. Proc. Natl. Acad. Sci. 2009, 106, 16157-62; Ding et al. Nat. Commun. 2019, 10, 1336; Sugahara et al. Science 2010, 328, 1031-1035). We have now added “iRGD (CRGDKGPDC) is 9-amino acid cyclic peptide that can bind to α V β 3 and α V β 5 integrins and further bind to neuropilin-1, and has shown good tumor-penetrating capability³⁶⁻³⁸” in the main text.

The purpose of Dox-iRGD is to kill a fraction of tumor cells and facilitate the release of tumor antigens from dying tumor cells. Meanwhile the dendritic cells accumulating at the biomaterial scaffold can sample and present those generated tumor antigens, followed by T cell priming and tumor killing. Therefore, this material system, capable of accumulating large numbers of DCs for in situ antigen presentation and subsequent immunomodulation, functions as a cancer vaccine.

We now provide data that the gels loaded with Dox-iRGD alone (without GM-CSF and CpG) did not show any therapeutic benefit compared to the untreated group (Fig. S3, also shown below), indicating that controlled delivery of Dox alone is not the main effect of the system. See also our response to Point 5 below, in relation to the vaccine function of the system.

Figure S3. Pore-forming gels containing Dox-iRGD alone do not show therapeutic benefit against 4T1 tumors. (a) Timeframe of efficacy study. Gels were peritumorally injected when the tumors grew to ~6-7 mm. The drug dose is described in Dox equivalent. (b) Tumor growth profiles for each group (n=5).

3. Regarding the nomenclature as a vaccine: Indeed, in the authors own explanation:

We appreciate the reviewer’s perspective on this topic. However, the USA Center for Disease Control (CDC) defines “vaccine” as “a product that stimulates a person’s immune system to produce immunity to a specific disease, protecting the person from that disease. Vaccines are usually administered through needle injections, but can also be administered by mouth or sprayed into the nose.” - this is clearly for a lay audience, not scientific explanation.

The World Health Organization (WHO) defines vaccine as follows: “A vaccine is a biological preparation that improves immunity to a particular disease. A vaccine typically contains an agent that resembles a disease-causing microorganism, and is often made from weakened or killed forms

of the microbe, its toxins or one of its surface proteins. The agent stimulates the body's immune system to recognize the agent as foreign, destroy it, and "remember" it, so that the immune system can more easily recognize and destroy any of these microorganisms that it later encounters.”- This clearly is an explanation of antigen(s). The technology proposed here utilizes doxorubicin to kill tumor cells, in situ, releasing antigen. So this is no more of a vaccine than administering systemic doxorubicin. I am actually quite surprised the authors chose to argue this, particularly from this direction. This aspect makes me concerned.

Response: We thank the Reviewer for these comments. We have now changed the title to ‘Biomaterial-Based In Situ Cancer Vaccine to Treat Poorly Immunogenic Tumors’, and have also avoided the use of ‘antigen free’ throughout the manuscript.

4. There are other serious misnomers/instances where the data do not support conclusions, for example in figure title claims. For instance, one egregious example: Supplemental Figure S7: Pore-forming gels containing GM-CSF, Dox-iRGD and CpG generate potent systemic tumor-specific CTL responses. Nothing in this figure shows anti-tumor immune responses, it simply shows interferon gamma production by T cells, and does not demonstrate that they are anti-tumor T cells. The same problem exists with the Figure S8 title.

Response: We apologize if this was not clear, and have modified our text to clarify that the IFN- γ data in Fig. 4d-e, Fig. S8 (former Fig. S7), and Fig. S9 (former Fig. S8) indicate tumor-specific T cell responses. In this analysis, cells were isolated from spleens and lymph nodes of vaccinated animals, and cocultured with 4T1 cells that were pre-seeded in the 96-well plates. These pre-seeded 4T1 cells provided stimulus for activating CD8⁺ and CD4⁺ T cells and facilitated their IFN- γ expression. In the negative controls, i.e. without 4T1 cells, the IFN- γ expression was minimal, validating the tumor-specific T cell responses. This is a routine assay to demonstrate tumor-specific cytotoxic T lymphocyte responses.

5. Finally, I could not find anywhere where the same experimental design was conducted in, for instance a RAG deficient or nude balb/c mouse to demonstrate that the gel 'vaccine' was actually slowing tumor growth due to immunological effects.

Response: We have now supplemented the data with a 4T1 tumor efficacy study in athymic nude mice (Fig. S23, also shown below). The gel vaccine failed to show any therapeutic benefit against 4T1 tumors in athymic nude mice. These data, combined with Fig. S3, in which a gel with dox-iRGD alone showed no therapeutic benefit over untreated mice, indicate that the therapeutic efficacy observed with the gel vaccine is not solely due to the contribution of the chemotherapy.

Figure S23. In situ gel vaccine does not show therapeutic benefit against 4T1 tumors in athymic nude mice. (a) Time frame of the animal study. Gels were loaded with GM-CSF, Dox-IRGD (200 μ g) and CpG (100 μ g). Athymic NU/J mice were used. (b) Tumor growth profiles for each group. (c) Kaplan-Meier plots for overall survival of all groups.

We also added the following text to the manuscript: “It is noteworthy that the gel vaccine did not show any therapeutic benefit in athymic nude mice with a compromised immune system (Fig. S23).”

6. The immunologic memory experiments in Figure 7, I truly cannot begin to understand what the authors' hypothesis is for why this would be generating immunologic memory when implanted locally, after tumors had been resected, if it is functioning as a 'vaccine'. More likely, the tumor is probably incompletely removed - i.e. residual tumor cells as stated by the authors - due to the fact that they are not mouse surgeons. How that hypothesis makes the model more clinically relevant is difficult to understand, since all positive margin resections are repeat surgery clinically; this is a very infrequent occasion. Overall, what is going on with systemic immunity is not clear.

Response: In the tumor resection studies, the tumor tissue/cells are removed as much as possible under a microscope. Nevertheless, 4T1 is such an aggressive tumor model that a minimal amount of residual tumor cells can result in tumor recurrence and further metastases. As the reviewer suggests, and we indicated in the previous version of the manuscript, our in situ gel vaccine can take advantage of the remaining tumor cells by inducing their death, leading to the accumulated dendritic cells first sampling locally and then presenting tumor antigens in lymph nodes to prime antigen-specific T cells.

Regarding how well tumors are resected in the clinic, we agree with the Reviewer that positive margins around breast tumors are resected in general. Given that ~20% of breast cancer patients ultimately relapse (Castaño, Z., et al. *Nat Cell Biol* 2018, 20, 1084–1097), and TNBCs in particular have the highest recurrence rates out of breast cancer subtypes (Cheng, L. et al. *Cancer Epidemiol*

Biomark Prev. 2012, 21, 800-809), we have considered several contexts in which our approach may be appreciated in the clinic. We have modified our text to include this relevant discussion. First, in the case of breast-conserving surgery, the removal of all tumor cells in the breast tissues is difficult, and likely a cause for tumor recurrence.⁵⁷ Second, for patients who already have metastases in distant tissues at the time of surgery, even in the case of mastectomy, tumor recurrence at the surgical site is not rare.⁵⁸ Our in situ gel vaccine can take advantage of any remaining or recurring tumor cells to boost the immune responses against both primary and metastatic cancers. Third, our approach is applicable to tumors that cannot be surgically resected or can only be partially resected in the clinic.

It is relevant to note that this same type of model (treatment at the site of primary tumor resection) has been utilized in a number of previous immunotherapy publications:

- (1) Piranlioglu R. et al. Primary tumor-induced immunity eradicates disseminated tumor cells in syngeneic mouse model. *Nature Communications*. 2019;10(1):1430.
- (2) Wang T. et al. A cancer vaccine-mediated postoperative immunotherapy for recurrent and metastatic tumors. *Nature Communications*. 2018;9(1):1532.
- (3) Zheng D. W. et al. A vaccine-based nanosystem for initiating innate immunity and improving tumor immunotherapy. *Nature Communications*. 2020;11(1):1985.
- (4) Ghochikyan A. et al. Primary 4T1 tumor resection provides critical "window of opportunity" for immunotherapy. *Clinical & experimental metastasis*. 2014;31(2):185-98. Epub 2013/10/08.
- (5) Song C. et al. Syringeable immunotherapeutic nanogel reshapes tumor microenvironment and prevents tumor metastasis and recurrence. *Nature Communications*. 2019;10(1):3745.

REVIEWERS' COMMENTS

Reviewer #2 (Remarks to the Author):

These authors have now satisfied all of my original and secondary concerns and I see no reason why the manuscript cannot be published in its present form.

Reviewer #4 (Remarks to the Author):

My critiques have been addressed, though I continue to be not at peace with the use of the word "vaccine". The words antigen-free were not really the primary source of contention or dissent among the reviewers.

Reviewer #2 (Remarks to the Author):

These authors have now satisfied all of my original and secondary concerns and I see no reason why the manuscript cannot be published in its present form.

Response: We appreciate the reviewer's positive comments.

Reviewer #4 (Remarks to the Author):

My critiques have been addressed, though I continue to be not at peace with the use of the word "vaccine". The words antigen-free were not really the primary source of contention or dissent among the reviewers.

Response: We thank the reviewer for their positive feedback. We have now changed the title to "Biomaterial-Based Scaffold for In Situ Chemo-Immunotherapy to Treat Poorly Immunogenic Tumors". All references to the "antigen-free vaccine" have been removed throughout the paper.